



# Chemical composition and source apportionment of atmospheric aerosols on the Namibian coast

**Danitza Klopper[1], Paola Formenti[2], Andreas Namwoonde[3], Mathieu Cazaunau[2], Servanne Chevaillier[2], Anaïs Feron[2], Cécile Gaimoz[2], Patrick Hease[2], Fadi Lahmidi[2], Cécile Mirande-Bret[2], Sylvain Triquet[2], Zirui Zeng[2] and Stuart J. Piketh[1]**

[1] North-West University, School for Geo- and Spatial Sciences, Potchefstroom, South Africa

[2] Laboratoire Interuniversitaire des Systèmes Atmosphériques (LISA), UMR CNRS 7583, Université Paris-Est-Créteil, Université de Paris, Institut Pierre Simon Laplace, Créteil, France

[3] SANUMARC, University of Namibia, Henties Bay, Namibia

**Corresponding author:** paola.formenti@lisa.u-pec.fr

**Abstract**

The chemical composition of aerosols is of particular importance to assess their interactions with radiation, clouds and trace gases in the atmosphere, and consequently their effects on air quality and the regional climate. In this study, we present the results of the first long-term dataset of the aerosol chemical composition at an observatory on the coast of Namibia, facing the southeast Atlantic Ocean. Aerosol samples in the mass fraction of particles smaller than 10 μm in aerodynamic diameter ($PM_{10}$) were collected during 26 weeks between 2016 and 2017 at the ground-based Henties Bay Aerosol Observatory (HBAO; 22°6'S, 14°30'E, 30 m above mean sea level). The resulting 385 filter samples were analysed by X-ray fluorescence and ion-chromatography for 24 inorganic elements and 15 water-soluble ions.

Statistical analysis by positive matrix factorization and back-trajectory modelling identified five major sources, sea salt (mass concentration: 70.8 ± 0.2%), marine biogenic (13.5 ± 0.8%), mineral dust (9.9 ± 0.1%), secondary products (3.2 ± 1.0%) and heavy metals (2.3 ± 2.5%). While the contribution of sea salt aerosol was persistent, as the dominant wind direction was south-westerly and westerly from the open ocean, the occurrence of mineral dust was episodic and coincided with high wind speeds from the south-southeast and the north-northwest, along the coastline. Concentrations of heavy metals measured at HBAO were higher than reported in the literature from measurements over the open ocean. The heavy metals (V, Cr, Nd and Mn) measured at the site were attributed to mining activities and the combustion of heavy fuels in commercial ship traffic across the Cape of Good Hope sea route. Fluoride concentrations up to 25 μg m$^{-3}$





were measured, as in heavily polluted areas in China. This is surprising and a worrisome result that has
profound health implications and deserves further investigation. Although no clear signature for biomass
burning could be determined, the source of secondary products identified by PMF was described by a
mixture of aerosols typically emitted by biomass burning, but also by other biogenic activities. Episodic
contributions with moderate correlations between $NO_3^-$, $nss-SO_4^{2-}$ (higher than 2 µg m$^{-3}$) and $nss-K^+$, were
observed, further indicative of the potential for an episodic source of biomass burning.
Sea salt accounted for up to 57% of the measured mass concentrations of $SO_4^{2-}$ and the non-sea salt fraction
contributed mainly to the secondary product and marine biogenic sources identified by PMF. The marine
biogenic contribution is attributed to efficient oxidation in the moist marine atmosphere of sulphur-
containing gas-phase emitted by marine phytoplankton in the fertile waters offshore in the Benguela
Upwelling System.
The data presented in this paper provide first-ever information on the temporal variability of aerosol
concentrations in the Namibian marine boundary layer and the links to meteorological conditions shaping
the transport patterns of aerosols from different sources. This data can be used to provide context for
intensive observations in the area.
**Keywords:** aerosols, chemical composition, transport, Namibia, positive matrix factorisation

## 1.  Introduction

Atmospheric aerosol particles are emitted from both natural and anthropogenic sources. Depending on
their chemical and physical characteristics, airborne aerosol particles modify the Earth's radiative budget
by scattering and absorbing solar and terrestrial radiation and by altering cloud lifetime and microphysical-
and optical properties (Seinfeld and Pandis, 2006). The variability in their source distribution and short
lifetime in the atmosphere (typically less than 10 days for particles below 1 µm in diameter, and shorter for
larger particles) results in an uneven horizontal and vertical spatial distribution of concentrations and
physicochemical properties (Seinfeld and Pandis, 2006). As a consequence, their effects on regional
atmospheric dynamics and processes are unevenly spread and constantly changing, in stark contrast to the
long-lived greenhouse gases which are well-distributed around the globe (Boucher, 2013).
The Namibian coast, and more generally the southeast Atlantic region of southern Africa, is amongst the
global areas of interest to study aerosols and their role on Earth's climate (De Graaf et al., 2014; Muhlbauer
et al., 2014; Painemal et al., 2014a, 2014b, 2014c; Wilcox, 2010; Zuidema et al., 2009). Local meteorological
conditions in this coastal desert environment are sustained by the effect of cold ocean currents in the
Benguela Upwelling System (BUS), one of the strongest oceanic upwelling systems in the world, with very
low sea surface temperatures all year round, reaching a minimum in the austral winter (Cole and



Villacastin, 2000; Nelson and Hutchings, 1983). This has a stabilising effect on the lower troposphere,
resulting in the formation of a semi-permanent stratocumulus (Sc) cloud deck extending between 10–30°S,
10°W–10°E, that tops the marine boundary layer at ~850 hPa (Muhlbauer et al., 2014; Wood, 2015) and is
of global significance for Earth's radiation budget (Klein and Hartmann, 1993; Johnson et al., 2004;
Muhlbauer et al, 2014; Wood, 2015).
The region is also known for its high marine phyto- and zoo-plankton specifically in the northern BUS
(Louw et al., 2016). The marine biogenic activity results in the release of gaseous compounds containing
sulphur (dimethylsulphide (DMS), $SO_2$, $H_2S$...) to the atmosphere (Andreae et al., 1994), whose oxidation,
particularly in this marine environment, could produce new aerosol particles contributing to the cloud
droplet number concentration of the Sc clouds (Charlson et al., 1987; Andreae et al., 1995). The region is
also known for the seasonal transport above the Sc of optically-thick and wide-spread smoke layers of
biomass burning aerosols emitted from forest fires in southern Africa in the austral dry season (August to
October; Lindesay et al., 1996; Swap et al., 2003).
Despite their relevance, very limited research has been conducted to assess the seasonal cycle and long-
term variability of the aerosol mass concentration and chemical composition in the region (Andreae et al.,
1995; Annegarn et al., 1983; Dansie et al., 2017; Eltayeb et al., 1993; Formenti et al., 1999, 2003b; 2018;
Zorn et al., 2008). To fill this gap, the long-term surface monitoring Henties Bay Aerosol Observatory
(HBAO) was established in 2012 on the campus of the University of Namibia's Sam Nujoma Marine and
Coastal Resources Research Centre (SANUMARC), along the Namibian coast (22°S; 14°E). HBAO faces the
open ocean in an arid environment, far from major point sources of pollution. Episodically through the year,
and seasonally between April to end of July, the station is affected by polluted air masses containing light-
absorbing aerosols, mostly from vegetation burning (Formenti et al., 2018).
In this paper, we present the results of the very first long-term measurements of aerosol elemental and
water-soluble ionic composition from the analysis of filter samples in the mass fraction of particles smaller
than 10 μm in aerodynamic diameter ($PM_{10}$ fraction) that were collected during 26 non-consecutive
sampling weeks in 2016 and 2017. The paper looks into the temporal variability of measured elemental
and water-soluble ionic concentrations and yields the first source apportionment to the $PM_{10}$ loading,
which is linked to the synoptic pathways of air mass transport.
The research presented in this study is relevant to the recent intensive observational efforts that took place
in Namibia in 2016 and 2017 (Zuidema et al., 2016). Specifically, it provides the long-term context to the
intensive filter sampling that was conducted in Henties Bay as part of the Aerosols, RadiatiOn and CLOuds
in southern Africa (AEROCLO-sA) project (Formenti et al., 2019).



## 2. Experimental methods

The HBAO station of Henties Bay, Namibia (22.09°S, 14.26°E; 30 m above mean sea level (amsl) http://www.hbao.cnrs.fr/, last access: 22 May 2019) is situated 100 m from the shoreline and is surrounded by an arid environment with little to no vegetation, as shown in Figure 1. Henties Bay is located approximatively 100 km north of Walvis Bay, the largest commercial harbour of Namibia (Namport, 2018). Formenti et al. (2018) showed that the location can be considered as baseline for a large part of the year (August to late April). However, from May to end of July it is impacted by the synoptic transport of light-absorbing aerosols mostly from vegetation burning in southern Africa and possibly but episodically by anthropogenic sources, such as heavy-fuel combustion by commercial ships travelling along the coast, especially along the Cape of Good Hope sea route (eg. Chance et al., 2015; Tournadre, 2014; Zhang et al., 2010).

### 2.1 Aerosol filter sampling and analysis

An automated sequential air sampler (model Partisol Plus 2025i, Thermo Fisher Scientific, Waltham, MA USA) was used to collect aerosol particles on 47-mm Whatman Nucleopore polycarbonate filters (1-μm pore size). Air was sampled at a flow rate of 1 $m^3$ $h^{-1}$ through a certified inlet (Rupprecht and Patashnick, Albany, New York, USA) located on the rooftop terrace above the instrument, and collecting aerosols particles of aerodynamic diameter lower than 10 μm ($PM_{10}$ fraction).

Individual filter samples were collected for 9 hours during the day (from 9 h to 18 h UTC) and during the night (from 21 h to 6 h UTC) on an intermittent week on/week off schedule. One blank sample per time series was collected. The whole dataset consisted of 385 samples.

Elemental concentrations of 24 elements (Na, Mg, Al, Si, P, S, Cl, K, Ca, Ti, V, Cr, Mn, Fe, Co, Ni, Cu, Zn, As, Sr, Pb, Nd, Cd, Ba) were obtained at LISA by wavelength-dispersive X-ray fluorescence (WD-XRF) using a PW-2404 spectrometer (Panalytical, Almelo, Netherlands), according to the protocol previously described by Denjean et al. (2016). The relative analytical uncertainty on the measured atmospheric concentrations (expressed in ng $m^{-3}$) is 5%, equal to the percent error on the certified mono- and bi-elemental standard concentrations (Micromatter Inc., Surrey, Canada) used for calibration of the XRF apparatus. Concentrations of elements from Na to Ca were corrected for the self-attenuation effects in large particles as proposed by Formenti et al. (2010).

The concentrations of 16 water-soluble ions ($F^-$, propionate, formate, acetate, methanesulfonic acid (MSA), $Cl^-$, $Br^-$, $NO_3^-$, $PO_4^{3-}$, $SO_4^{2-}$, oxalate, $Na^+$, $NH_4^+$, $K^+$, $Ca^{2+}$ and $Mg^{2+}$) were obtained at LISA by ion chromatography (IC) with a Metrohm IC 850 device (injection loop of 100 μl). For anionic species, the IC was equipped with MetrosepA supp 7 (250/4.0mm) column associated with a MetrosepA supp 7 guard pre-column heated at





45°C. For simultaneous separation of inorganic and short-chain organic anions, elution has been realised
with the following elution gradient (eluent weak:$Na_2CO_3$/$NaHCO_3$ (0.28/0.1mM) and eluent strong:
$Na_2CO_3$/$NaHCO_3$ (28/10mM): 100% eluent weak from 0 to 23.5 minutes; then 15% eluant strong from 23.5
to 52 minutes and 100% eluent weak to finish. The elution flow rate was 0.8 ml min$^{-1}$. For cationic species,
IC has been equipped with a Metrosep C4 (250/4.0mm) column associated to a Metrosep C4 guard column
heated at 30°C. Elution has been realised with an eluant composed with 0.7 mM of dipicolinic acid and
1.7mM of nitric acid. The elution flow rate was 0.9 ml min$^{-1}$. The uncertainty of water-soluble ionic
concentrations (also expressed in ng m$^{-3}$) is within 5%, the maximum uncertainty obtained during
calibration by standard certified mono- and multi-ionic solutions.
For each chemical species, the minimum quantification limit (MQL) was calculated as 10 times the square
root of the standard deviation of the concentration of laboratory blank samples, corresponding to filter
membranes prepared as actual samples but stored and analysed without exposure to external air. Only
values above MQL are included in further analyses.
A quality-check assessment of the analysis was performed by comparing the concentrations of Cl, Mg, K, Ca,
Na and $SO_4^{2-}$/S measured by IC and XRF (Figure S1). The comparison revealed a good linear correlation
between the two datasets, with the coefficient of determination ($R^2$) exceeding 0.85 for all the elements.
However, some differences in the slopes of the linear correlations are observed when comparing the 2016
and 2017 datasets for Cl$^-$/Cl, Na$^+$/Na, and Mg$^{2+}$/Mg. Slopes were 1.3 ± 0.1 (2016) and 1.0 ± 0.1 (2017), 1.3
± 0.1 (2016) and 0.9 ± 0.1 (2017), and 1.3 ± 0.1 (2016) and 1.5 ± 0.2 (2017) for Cl$^-$/Cl, Na$^+$/Na, and Mg$^{2+}$/Mg,
respectively. Conversely, no annual dependence was observed in the slopes of the linear correlations for
Ca$^{2+}$/Ca (0.8 ± 0.1), K$^+$/K (0.8 ± 0.1) and $SO_4^{2-}$/S (2.7 ± 0.3). These values are in general terms consistent
with expectations that these elements, mostly but not exclusively composing sea salt, should be
predominantly soluble in water. However, ratios higher than unity are obtained for Cl$^-$/Cl and Na$^+$/Na for
2016, and Mg$^{2+}$/Mg for both years. Although no specific sampling nor analytical problems were found, the
further comparison of their proportions to those expected for seawater (Seinfeld and Pandis, 2006)
suggested to discard the XRF results and only use the values obtained by IC for those three elements. For
Ca$^{2+}$/Ca, K$^+$/K and $SO_4^{2-}$/S, ratios are consistent with previous observations in marine environments
impacted by mineral dust (Formenti et al., 2003a).
**2.2     Local winds, air mass trajectories and synoptic meteorology**
Local wind speed and direction were measured with two anemometers also located on the rooftop of HBAO,
first, a Campbell Scientific 05103, replaced with a Vaisala WXT530 from September 2017 onwards.
Measurements were stored as 5-minute averages. Wind data was available for all of 2016 and 55% of the
aerosol sampling periods in 2017 (no wind data were available during 19 – 26 May and 7 – 14 July 2017).





The NOAA Hybrid Single-Particle Lagrangian Integrated Trajectory (HYSPLIT) model (Stein et al., 2015)
was used to evaluate the origin and transport pathway of air masses to HBAO. Seventy-two-hour back-
trajectories were run every hour for each 9-hour long filter sampling period starting at a height of 250 m
above ground level (agl), which effectively models transport into the marine boundary layer (MBL, with a
minimum height of ~500 m over the BUS; Preston-Whyte et al., 1977). This choice also considered the
model vertical resolution (23 levels throughout the atmospheric column). The first model vertical level is
at 1000 hPa (approximately 110 m amsl) and the next is at 975 hPa (approximately 300 m amsl). The Global
Data Assimilation System (GDAS) reanalysis dataset with a 1° x 1° resolution, provided by the National
Centre for Environmental Prediction (NCEP), was used. This was preferred to the 0.5° x 0.5° resolution
dataset where the vertical velocity is absent and has to be calculated from the divergence, introducing
uncertainties into the model. Trajectories were run through the Rstudio interface using the
rich_iannone/splitR (available from https://github.com/rich-iannone/ SplitR) and Openair (Carslaw and
Ropkins, 2017) packages from the open-source libraries.
As a complement, publicly available daily synoptic charts provided by the South African Weather Service
(SAWS, www.weathersa.co.za/home/historicalsynoptic; last accessed 20/02/2020) were analysed for the
synoptic-scale induced flow.

### 3.   Source identification and apportionment

The identification of the origin of the aerosols, complementary to the analysis of the air mass back
trajectories and local wind speed and direction, was undertaken by examining the temporal correlations of
the elemental and ionic concentrations to known tracers and positive matrix factorisation.

### 3.1   Ratios to unique tracers

The identification and quantification of the aerosol types contributing to the total particle load at HBAO
were done by investigating the linear correlation of measured elemental and ionic concentrations and their
mass ratios to unique tracers of the atmospheric particulate matter emission sources expected in the
region. These are:
● Sea salt aerosols, traced by $Na^+$ constituting 30.6 % of the aerosol mass in seawater (Seinfeld and
187       Pandis, 2006);
● Marine biogenic emissions during the life cycle of marine phytoplankton in the BUS (Nelson and
Hutchings, 1983) and traced by the concentrations of particulate MSA, a unique product of the
oxidation of gaseous DMS (Seinfeld and Pandis, 2006);
● Wind-blown mineral dust liberated from the surface of pans and ephemeral river valleys (Annegarn
et al., 1983; Eltayeb et al., 1993; Heine and Völkel, 2010; Dansie et al., 2017), but also during /road
construction and mining activities (KPMG, 2014). Mineral dust is traced by elemental aluminium,




representing aluminosilicate minerals and contributing on average 8.13% of the global crustal rock
composition by mass (Seinfeld and Pandis, 2006), and by the non-sea salt (nss) fraction of $Ca^{2+}$ to
represent calcium carbonate. This is justified by the specific mineralogy of Namibian soils which
are enriched in gypsum ($CaSO_4OH$) and calcite ($CaCO_3$), and presenting calcium content higher than
the world average (Annegarn et al., 1983; Eltayeb et al., 1993). The apportionment of the sea-salt
(ss) and non-sea salt (nss) $Ca^{2+}$ fractions was done using the nominal mass ratio of $Ca^{2+}/Na^+$ in
seawater (0.021; Seinfeld and Pandis, 2006). The evaluation of the mass concentration of calcium
carbonate was done by multiplying the measured nss-$Ca^{2+}$ mass concentration by the $CaCO_3/Ca$
mass ratio of 2.5;
● Elements such as Ni, V, Pb, Cu, Zn to trace heavy-oil combustion from industry and commercial
shipping (Becagli et al., 2017; Johansson et al., 2017; Sinha et al., 2003; Vouk and Piver, 1983) as
well as mining activities (Ettler et al., 2011; Kříbek et al., 2018; Soto-Viruet, 2015), and;
● nss-$K^+$, calculated from measured $K^+$ assuming the mass ratio $K^+/Na^+$ of 0.036 as in seawater
(Seinfeld and Pandis, 2006), to trace seasonal transport of biomass burning aerosols (Andreae et
al., 1998; Andreae and Merlet, 2001).
**3.2 Positive matrix factorisation**
Positive matrix factorization (PMF) multivariate statistical methods are widely used to identify source
profiles and explore source–receptor relationships using the trace element compositions of atmospheric
aerosols (e.g., Schembari et al., 2014). The PMF uses weighted least-squares factor analysis to deconvolute
the matrix of observed values (X) as $X = G \times F + E$, where $G$ and $F$ are the matrices representing the factor
scores and factor loadings, respectively, and $E$ is the matrix of residuals equal to the difference between
observed and predicted values (Paatero and Tapper, 1994; Paatero et al., 2014).
In this paper, the multivariate PMF statistical analysis was conducted with the EPA (Environmental
Protection Agency) PMF version 5.0 (Norris et al., 2014). The XRF and IC datasets were combined by
retaining only elements/ions measured above the MQL in more than 70 samples (that is, at least in 20% of
the collected values). This criterion excluded Ba, $Br^-$, $PO_4^-$ and $Mn^{2+}$. Occasional missing values in the
retained elements/ions were replaced by the species median value, as recommended in Norris et al.,
(2014). Uncertainties for all missing values in the dataset were replaced by a dummy value (99999) to
ensure that these samples do not skew the model fit (Norris et al., 2014). According to section 5.2.1, the
water-soluble ionic form instead of the elemental form was retained for elemental Mg, Na, Cl, K, Ca and S.
$SO_4^{2-}$, $K^+$ and $Ca^{2+}$ were included as their ss- and nss- fractions. The final matrix was made up of 385
observations of 33 chemical species. By comparing the signal to noise ratio, the nss-K+ was the only species
in the matrix marked as a "weakBased on the temporal correlation, the PMF analysis resolves the chemical





dataset into a user-specified number of solutions (sources). No completely objective criterion exists for
selecting the number of sources and so the model was run considering potential solutions of three to seven
sources. Each of these models were run 100 times using randomised seeds. For each of these runs, the
robustness-of-fit was compared and the estimation of the error range of each solution was done by
displacing chemical species in each modelled source and testing the rotational ambiguity of the solutions
(Norris et al., 2014; Paatero et al., 2014). *Fpeak* rotations with strengths between -0.5 and 1.5 were tested
to further optimise the source solutions. An *Fpeak* strength of 0.5 was used for the final solution. From these
analyses, the solution with the smallest uncertainties and greatest stability was selected as the most
physically meaningful and representative solution.

## 236    4.   Results and discussion

### 237    4.1    Meteorological conditions during sampling

The characteristic synoptic circulation patterns identified over the west coast of southern Africa that are
significant for this study include continental-anticyclonic circulation., the southeast Atlantic anticyclone,
west coast troughs and barotropic easterly waves, transient baroclinic westerly waves and coastal low-
pressure systems (Tyson and Preston-Whyte, 2014). Formenti et al. (2018), found that anticyclonic
circulation, both in the form of the South Atlantic anticyclone and the continental anticyclone, is the most
persistent circulation patterns over the west coast of Namibia.
Figure 2 shows weekly composite maps of calculated air mass back trajectories (their gridded frequency
plot is shown in Figure S2). Southerly and south-westerly transport occurred year-round and easterly
transport mainly occurred during late autumn (May), winter (June, July and August) and early spring
(September, October and November). Large scale north-easterly air mass transport towards HBAO was
restricted to the austral autumn and winter when continental anticyclonic flow dominated the circulation
patterns in the lower and mid-troposphere. The majority of air masses arriving in the MBL are of marine
origin from the southern and south-eastern Atlantic and show the transport of marine air masses toward
the subcontinent, divergence at the escarpment and southerly flow, induced along the coast. Most of the air
masses were transported over coastal waters offshore and along the west coast of South Africa and Namibia
and just inland to the north-northeast of HBAO from the sub-continent. Continental plumes arriving at
HBAO are transported easterly between 15° and 22°S and from as far as 36°E.
Emissions along these preferred pathways may be of great significance in shaping the regional aerosol
background. Some of the known transport regimes are associated with mid-tropospheric easterly winds,
responsible for transport off the subcontinent (Swap et al., 1996; Tyson et al., 1996). To the north of HBAO,
Adebiyi and Zuidema (2016) observed continental plumes transported off the coast, especially under
anticyclonic circulation over the subcontinent and the southeast Atlantic Ocean. Tlhalerwa et al. (2012)





found berg winds, an easterly perturbation, to be the main agents of aerosol transport and deposition off
the coast at Luderitz, around 500 km south of HBAO, and easterly winds in the boundary layer may
transport dust from the subcontinent into the ocean.
The weekly and hourly variability of local surface winds is illustrated in Figures 2 and 3, respectively. On
average the wind is characterised by low speeds during the daytime ($4.7 \pm 2.2$ m s$^{-1}$ with only 0.3% calm)
and at night ($3.3 \pm 2.1$ m s$^{-1}$ with 0.6% calm conditions). The low wind speeds are typical for regions
frequently experience anticyclonic circulation. The highest wind speeds were recorded for southerly winds
which were persistent throughout the sampling period, except during January 2017 (Figure 2). The highest
wind speed was recorded in the austral spring in both years and reached a maximum of 18.9 m s$^{-1}$ in the
week of 13–20 November 2017.
Another feature that is promoted by anticyclonic flow is thermally-induced land and sea breezes. Sea
breezes were a common daytime occurrence at HBAO. The sea breeze is typically characterised by
southerly and south-westerly winds. The wind direction is partly a function of the shape of the coastline at
Henties Bay and the overlying gradient flow.  The daytime land breeze was not observed as frequently as
the onshore sea breeze flows. This supports the conclusion that the mechanisms for onshore flow are a
combination of local and large-scale circulation. ENE and northerly winds were seen in July 2016 reaching
a maximum speed of 13 m s$^{-1}$ (mean wind speed of $4.5 \pm 2.2$ m s$^{-1}$ for the week of 19-26 July 2016). These
are the land breezes that are also most likely to develop on clear stable nights. The northerly flow, in
particular, occurred in the early evening and mid-morning (Figure 4), with no seasonal dependence.
Overall, it is important to note that the sea-breeze winds during the day are well defined in the data. At
night the land breeze is much less important at Henties Bay than one might expect at a coastal site. This is
almost certainly driven by the small thermal gradient that exists between the ocean and land temperatures
at night. In the absence of a well-defined gradient, the land breeze does not develop on most nights.
Direct westerly winds occur less frequently at the site. The winds could be observed during the day and the
night indicating that they are not exclusively established as sea breeze cells. The wind speeds for westerly
flow conditions never exceeded 6 m s$^{-1}$.
Easterly winds were only observed during the warmer months (January to March and September to
December, Figure 3), and during the night-time sampling periods (21 to 9 UTC), when their speeds
remained below 4 m s$^{-1}$ (Figure 4). This local circulation is driven by easterly wave or tropical easterly
circulation that moves southward during the summer months.





### 4.2 Variability and apportionment of measured concentrations

A summary of the measured elemental and water-soluble mass concentrations (arithmetic mean, standard
deviation and range of variability) at HBAO during 2016 and 2017 are provided in Table 1. The time series
of the mass concentrations of the source tracers discussed in section 3.1 are shown in Figure 5.

#### 4.2.1 Sea salt

As expected, the major components of sea salt aerosols ($Cl^-$, $Na^+$, $Mg^{2+}$ and $K^+$) were sampled in high
concentrations (up to 76, 53, 5.6 and 2.0 µg m$^{-3}$, respectively) throughout the sampling periods. Their time
variability, illustrated in Figure 5 by the example of $Na^+$, was very similar and characterised by a significant
continuous background that could be represented by a 10-point moving average (that is, 90 hours). The
calculated mean background concentration was 10.1 ± 3.6 µg m$^{-3}$. No seasonal cycle was evident due to the
dominance of southerly and south-westerly winds transporting marine air masses onshore (Figure 3).
Table 2 shows the mass ratios of $Cl^-$, $Mg^{2+}$, $K^+$, $Ca^{2+}$, $F^-$ and $SO_4^{2-}$ to $Na^+$ for 2016 and 2017 calculated as the
slopes of their linear regression lines, and evaluated by the coefficient of determination ($R^2$). The
experimental values were compared with average ratios in seawater (Seinfeld and Pandis, 2006). The
average $Cl^-/Na^+$ mass ratio was 1.35 ± 0.11 in 2016 and 1.34 ± 0.11 in 2017, lower by 25% than the value
expected in seawater of 1.8. This difference has previously been reported in fresh sea salt in acidic marine
environments (e.g., Zhang et al., 2010), and is attributed to $Cl^-$ depletion via reactions between NaCl and
sulfuric- and nitric acids. A very good correlation was observed between the ratios of $Mg^{2+}$(0.12 ± 0.01)
and $K^+$(0.04 ± 0.01) to $Na^+$ in this data set and the value reported for sea water (Table 2) and, (Seinfeld and
Pandis, 2006). Conversely, the linear correlation between $Ca^{2+}$ and $Na^+$ (not shown) was less pronounced
($R^2$ = 0.61 and 0.42 in 2016 and 2017, respectively). The $Ca^{2+}/Na^+$ mass ratio was systematically higher
than in seawater (0.04), indicating the contribution of crustal calcium typical of the Namibian soils (see
section 4.2.2). Using the average seawater ratio, the mean sea-salt (ss) $Ca^{2+}$ concentration was estimated as
470 ± 360 ng m$^{-3}$ and 360 ± 210 ng m$^{-3}$ for 2016 and 2017, respectively. The mean non-sea-salt (nss) $Ca^{2+}$
concentration was 420 ± 520 and 270 ± 400 ng m$^{-3}$, respectively for the two years, representing 47% and
42% of the mean measured $Ca^{2+}$ concentrations. Similarly, for both 2016 and 2017, the ss and nss
components of $K^+$ were estimated as 367 ± 246 ng m$^{-3}$ and 44 ± 54ng m$^{-3}$ respectively, accounting for 89 %
and 11% of the $K^+$ mass. Finally, the $SO_4^{2-}/Na^+$ mass ratio for HBAO (0.36 ± 0.14) was higher than the
average mass ratio for seawater (0.25). The origin and significance of this excess are explored in section

319 4.2.6.

#### 4.2.2 Mineral dust

The time series of Al and nss-$Ca^{2+}$ (Figure 5) were investigated to identify the transport of airborne mineral
dust at Henties Bay. Mineral dust episodes were identified when the concentrations of those tracers



exceeded background values (modelled as the 10-point moving average) for a minimum of 3 consecutively
sampled filters. Similar time variability was observed for elemental Fe, Si, Ti and P (not shown). Overall, 19
episodes of mineral dust were identified (Table S1).
The mean mass concentration of elemental Al was 556 ± 643 ng m$^{-3}$ in 2016 and 446 ± 551 ng m$^{-3}$ in 2017,
while values peak as high as 4.7 μg m$^{-3}$ (Table 1). To the best of our knowledge, no other measurements of
Al are available in Namibia for comparison. Our nss-Ca$^{2+}$ mean (703 ± 644 ng m$^{-3}$ in 2016 and 428 ± 437 ng
m$^{-3}$ in 2017) is similar to the concentrations measured by Annegarn et al. (1983)(425 ng m$^{-3}$ and maximum
of 800 ng m$^{-3}$) in central Namibia at Gobabeb, in the Namib Desert, (23°45'S, 15°03'E). The annual mean Fe
concentration measured at HBAO (372 ± 480 ng m$^{-3}$ in 2016 and 338 ± 433 ng m$^{-3}$ in 2017) compares well
with the average of 246 ng m$^{-3}$ provided by Annegarn et al. (1983).
Table 3 shows the mass ratios for major components of mineral dust as well as some heavy metals (V and
Ni). Overall, Si, Fe, and Ti showed very good correlations to Al as expected for mineral dust ($R^2$ > 0.9). The
average mass ratio of Si/Al was 3.7± 1.0 in 2016 and 3.3 ± 0.9 in 2017, lower than the average values of 4
to 4.6 expected in global soils and crustal rock (Seinfeld and Pandis, 2006). This is attributed to the size-
fractionation during aeolian erosion of soils producing airborne dust. As a matter of fact, our average values
are consistent with those obtained for particles less than 10 μm in diameter by Eltayeb et al. (1993) at
Gobabeb. Our averages, generally higher than in mineral dust from north Africa (Formenti et al., 2014),
compare well with the value (3.4) reported by Caponi et al. (2017) for mineral dust aerosols generated in
a laboratory experiment from a soil collected to the northeast of HBAO. The average Fe/Al ratio was 0.7 ±
0.2 in 2016 and 0.8 ± 0.04 in 2017, lower than the ratio of 1 reported by Eltayeb et al. (1993). The same is
observed for the Ti/Al ratio, which was 0.07 ± 0.02 in 2016 and 0.06 ± 0.03 in 2017 at HBAO and
approximately 0.15 in Eltayeb et al. (1993).
The average nss-Ca$^{2+}$/Al ratio was 1.6 ± 0.7 in 2016 and 1.4 ± 0.7 in 2017. However, for the strongest dust
episodes (Al values higher than 1 μg m$^{-3}$) the ratio tended to 1 (Figure 6). This is in agreement with the
specific mineralogy of Namibian soils that are rich in limestone and gypsum (Annegarn et al., 1983; Eltayeb
et al., 1993). The mean Fe/nss-Ca$^{2+}$ ratio was 0.5 ± 0.2 in 2016 and 0.7 ± 0.2 in 2017, higher than the value
of 0.11 ± 0.10 reported by Caponi et al. (2017), pointing to the diversity in soil mineralogy, even at relatively
small spatial scales.
Figure 6 also shows the nss-K$^+$/Al ratios as a function of Al. As for nss-Ca$^{2+}$, values were spread but ranged
between 0.1 and 0.5 when Al concentrations exceeded 1 μg m$^{-3}$. These values are in agreement with those
for mineral dust sources in North Africa (Formenti et al., 2014).





The average phosphorus concentrations measured at HBAO were 11 ± 9 ng m⁻³ in 2016 and 14 ± 4 ng m⁻³
in 2017. Phosphorous was very well correlated with Al in 2016 ($R^2$ = 0.92) and only moderately correlated
in 2017 ($R^2$ = 0.66). The P/Al ratio averaged 0.03 ± 0.02 in 2016 and 0.05 ± 0.02 in 2017. As was observed
for the nss-Ca$^{2+}$/Al, the P/Al ratio tended to an asymptotic value of 0.02 when Al exceeded 1 μg m⁻³ (not
shown). This is significantly higher than reported by Formenti et al. (2003a) for the outflow of Saharan dust
to the North Atlantic Ocean (0.0070 ± 0.0004).
The mean concentrations of mineral dust elements Al, Fe, Ti and Si were higher for night-time sampling
between 21 and 06 UTC, and lower in the day (9 to 18 UTC). It follows that the local measurement of easterly
winds was only observed at night and in the early morning (Figure 4), consistent with these higher
concentrations in mineral dust elements.
### 4.2.3  Heavy metals
Vanadium and nickel are naturally occurring in mineral deposits in soils (Annegarn et al., 1983; Maier et
al., 2013), but they are also known tracers of heavy-fuel combustion, as reported in Becagli et al. (2013)
and references therein. Their average concentrations at HBAO were 9 ± 5 ng m⁻³ (2016) and 7 ± 6 ng m⁻³
(2017) for V, and 8 ± 7 ng m⁻³ (2016) and 7 ± 4 ng m⁻³ (2017) for Ni. These values are an order of magnitude
larger than measured over the open ocean by Chance et al. (2015) and comparable to those measured by
Isakson et al. (2001) at a Swedish harbour and by Becagli et al. (2012) in the central Mediterranean Sea
downwind of a major shipping route. In our study, V and Ni were relatively well correlated. Vanadium was
well correlated with Al when Al exceeded 1 μg m⁻³ ($R^2$ around 0.4), whereas no correlation between Ni and
Al were observed (Figure 6). Additionally, the correlation of V to Si, also used in the literature as a tracer of
mineral dust, was evident while moderate ($R^2$ around 0.4), while none was found for Ni. This differs from
what was reported by Becagli et al. (2012), who found that neither V nor Ni were correlated to Si. In our
dataset, both V/Si and Ni/Si ratios were enriched by a factor of 10 or more to reference values for the upper
continental crust (3.1×10⁻⁴ and 1.5×10⁻⁴ for V/Si and Ni/Si, respectively; Henderson and Henderson, 2009).
The highest V concentrations corresponded to south-south-easterly winds while the highest Ni
concentrations were found for southerly to westerly winds.
These facts, and the moderate to good correlations of V and Ni with Zn, Cu and Pb, suggest that V and Ni do
not necessarily have the same sources. Mining activities, likely in the Otavi mountain area (Boni et al.,
2007), should account for the high concentrations of V, with additional contributions from combustion of
heavy fuels, where V is present as an impurity (Isakson et al., 2001, and references therein; Vouk and Piver,
1983). On the contrary, combustion of heavy fuels seems to be the primary source of Ni. Indeed, HBAO is
located downwind one of the major ship tracks of southern Africa (Tournadre, 2014). The V/Ni ratio for





2016 is 1.7 ±1.1 and 2017 is 1.3 ± 1.3, lower than 2.8-2.9 reported for the central Mediterranean (Becagli
et al., 2017) and 4-5 reported for a harbour in Melilla, Spain (Viana et al., 2009).
Zn and Pb are also found as impurities in bulk fuels for ships (Isakson et al., 2001). The mean concentration
of Zn at HBAO (11 ± 9 ng m$^{-3}$) was about two orders of magnitude higher than over the southeast Atlantic
Ocean (Chance et al., 2015) and in desert air (Annegarn et al., 1983). Likewise, the mean Pb concentration
(75 ± 89 ng m$^{-3}$) was three orders of magnitude higher than reported by Chance et al. (2015) for soluble Pb
and comparable to values measured in the western Mediterranean by Denjean et al. (2016). Average
concentrations of Cu at HBAO were 8 ± 6 ng m$^{-3}$, an order of magnitude higher than measured in windblown
dust by Annegarn et al. (1983) in the central Namib but two orders of magnitude smaller than the average
measured by Lee et al. (1999) in highly polluted Hong Kong (125.1 ng m$^{-3}$). Ettler et al. (2011) showed that
copper ore mining and smelting operations in the Zambian Copperbelt are a significant source of potentially
bioavailable copper, that, unlike phosphorus, has been found to inhibit plankton growth in laboratory
studies (Paytan et al., 2009) and over the western Mediterranean (Jordi et al., 2012). Similar contamination
of topsoil was found by Kříbek et al. (2018) at operations in the Tsumeb mining district, Namibia (19°14'S,
17°43'E).

### 4.2.4  Fluoride

Atmospheric fluoride is primarily from fluorspar mining in the Okorusu Mine (20°3'S, 16°44'E), but very
likely also from the surface mining occurring approximately 20 km south of HBAO to provide gravel for the
construction of a major road between Swakopmund and Henties Bay which started late in 2015 (A.
Namwoonde, *pers. corr.*). Fluoride may be leached into groundwater from fluoride-rich soils throughout
the region (Wanke et al., 2015), which may then evaporate when exposed to the atmosphere.
One of the striking features of Table 1 is the high mean concentration of F$^-$ measured at HBAO (4.3 ± 4.0 μg
m$^{-3}$ in 2016 and 2.8 ± 2.5 μg m$^{-3}$ in 2017), with peak values as high as 25 μg m$^{-3}$. Those annual mean
concentrations were comparable to the mean 24-h fluoride concentrations measured between 1985 and
1990 over the South African Highveld by Scheifinger and Held (1997). The peak values at HBAO were
significantly higher than maxima reported by these authors and ranging between 1.4 and 2.9 μg m$^{-3}$. The
measured concentrations at HBAO were also comparable to those of heavily polluted areas in China (Feng
et al., 2003), and significantly higher than reported for Europe, even in the polluted Venice lagoon (Prodi et
al., 2009) or in areas nearby ceramic and glass factories (Calastrini et al., 1998). The mean F$^-$/Na$^+$ mass ratio
measured at HBAO was 0.39 ± 0.29 in 2016 and 0.32 ± 0.29 in 2017, enriched by several orders of
magnitude to average seawater composition (mass ratio 1.2x10$^{-4}$; Table 2). The very good correlation of F$^-$
with nss-Ca$^{2+}$ ($R^2$ equal to 0.76 in 2016 and to 0.84 in 2017) yielded a mean mass ratio of 6.4 and 5.8,
respectively, much higher than reported in groundwater, aerosols or precipitation in polluted





419 environments (Feng et al., 2003; Prodi et al., 2009). The highest F⁻ concentrations were associated with

420 south to the easterly winds, that is, from the subcontinent.

421 **4.2.5 Arsenic**

422 Inorganic arsenic in geologic formations may be released by mining operations or evaporated from soil and

423 groundwater (Gomez-Caminero et al., 2001). No correlation between As to Al nor nss-$Ca^{2+}$ was found.

424 Arsenic is also released by marine algae and plankton (Sanders and Windom, 1980; Shibata et al., 1996).

425 Again, no discernible correlation between As and MSA was found. Other sources of arsenic are biomass

426 burning, fossil fuel combustion and non-ferrous metal smelting operations (Ahoulé et al., 2015; Gomez-

427 Caminero et al., 2001). The arsenic concentrations at HBAO have a mean and standard deviation for 2016

428 of 22 ± 16 ng m⁻³ and 239 ± 344 ng m⁻³ in 2017. The mean for 2017 is skewed due to two sampling weeks

429 with very high concentrations in the order of those measured in rural and urban-industrial areas affected

430 by mining and smelting emission sources (Hedberg et al., 2005; Šerbula et al., 2010). For these two weeks,

431 the MSA concentrations were slightly higher than the mean reported in Table 1 (77 ± 33 ng m⁻³) as were

432 the Sr concentrations (146 ± 53 ng m⁻³). This may indicate contributions during these two weeks from

433 marine biogenics and/or from the Tsumeb smelter in the northeast of HBAO (KPMG, 2014).

434 **4.2.6 Sulphate and secondary aerosols**

435 The annual mean sulphate concentration measured at HBAO was 4.1 ± 2.6 µg m⁻³ in 2016 and 3.4 ± 1.4 µg

436 m⁻³ in 2017 (Table 4), higher than previously measured over the southern Atlantic and Pacific oceans

437 (Zhang et al., 2010) and comparable to springtime measurements in the Venice Lagoon (Prodi et al., 2009).

438 As shown in Formenti et al. (2019), the highest concentrations were measured in spring and autumn, while

439 minima occurred between May and August. $SO_4^{2-}$ and $Na^+$ showed good correlation ($R^2$ = 0.92 in 2016 and

440 0.83 in 2017, Table 2) but their mass ratios were higher than in seawater (0.36 ± 0.14 and 0.42 ± 0.23 in

441 2016 and 2017, respectively, compared to the expected mass ratio of 0.25; Seinfeld and Pandis, 2006). The

442 apportionment of the ss and nss fractions of $SO_4^{2-}$ was done using the nominal mass ratio of $SO_4^{2-}$/$Na^+$ in

443 seawater (0.25; Seinfeld and Pandis, 2006). As a result, up to 57% of the measured $SO_4^{2-}$ mass concentration

444 in the $PM_{10}$ fraction was attributed to sea salt aerosols, and the nss-component was of the order of 43%.

445 This is in agreement with previous observations in the south Atlantic Ocean (Andreae et al., 1995; Zhang et

446 al., 2010; Zorn et al., 2008). On the contrary, at the remote Brand se Baai site along the Atlantic coast of

447 South Africa (31.5°S 18°E), Formenti et al. (1999) reported than sea salt accounted for about 92% of the

448 total measured elemental sulphur concentrations.

449 The MSA concentrations measured at the site ranged between 10 and 230 ng m⁻³ (Table 1). The mean

450 annual concentration was 63 ± 39 ng m⁻³, three times higher than the mean value of 20 ± 20 ng m⁻³ (6.2 ±

451 4.2 ppt) reported by Andreae et al. (1995) over the open ocean along 19°S, and lower than in the southeast





Atlantic Ocean (Zhang et al., 2010; Table 4). As shown by Formenti et al. (2019), the MSA concentration at
HBAO displayed a clear seasonal cycle, with higher values in spring to summer, as previously observed in
the Southern Hemisphere ocean due to more efficient oxidation of DMS in warmer conditions (Ayers et al.,
1997; Huang et al., 2017). This also explains the highest mean concentrations of marine biogenic products
(MSA, As and nss-$SO_4^{2-}$) measured in the morning and the lowest at night. Springtime averages for MSA
were in the range of that measured by Huang et al. (2018) during a springtime cruise over the South Atlantic
and by Prodi et al. (2009) in the Venice Lagoon (Table 4).
The MSA/nss-$SO_4^{2-}$ ratio (Figure 7) displayed a large range of values (0.01 to 0.12), in agreement with
values in the literature at various geographical locations, especially in the southern Hemisphere (Table 4).
As pointed out by Formenti et al. (2019), the MSA/nss-$SO_4^{2-}$ values were higher in the austral summer and
spring and lower in the austral winter. This strong seasonal dependence is in agreement to that identified
by Ayers et al. (1986) for marine biogenic sulphur in the Southern Hemisphere and suggests that the highest
concentrations of nss-$SO_4^{2-}$ in the $PM_{10}$ (nss-$SO_4^{2-}$ larger than 2 µg m$^{-3}$) are not necessarily associated to
marine biogenic emissions. From measurements at the desert station of Gobabeb, in the Namib Desert,
Annegarn et al. (1983) found that only the fine mode of the bimodal distribution of sulphur aerosols, that
is, that bearing the lower mass concentrations, would be due to the oxidation of sulphur-containing gaseous
emissions during the marine phytoplankton life cycle.
Figure 7 also illustrates the $NH_4^+$/nss-$SO_4^{2-}$ mass ratio as a function of nss-$SO_4^{2-}$ mass concentrations. Both
in 2016 and 2017, the $NH_4^+$/nss-$SO_4^{2-}$ mass ratios were less variable than for MSA/nss-$SO_4^{2-}$. The annual
mean $NH_4^+$/nss-$SO_4^{2-}$ were 0.13 ± 0.10 in 2016 and 0.14 ± 0.08 in 2017. These values are consistent with
the mass ratio of 0.18 corresponding to ammonium bisulphate (($NH_4$)$HSO_4$). Although some losses of $NH_4^+$
due to conservation on-site and transport to the laboratory in France cannot be excluded, the measured
ratios are consistent with previous investigations in remote marine environments reported in Table 4,
including offshore southern Africa (Andreae et al., 1995; Quinn et al., 1998).
The average $NO_3^-$/nss-$SO_4^{2-}$ ratio at HBAO was of the order of 0.14, significantly smaller than reported by
Zhang et al. (2010) over the southeast Atlantic. Finally, poor correlation between nss-$SO_4^{2-}$ and nss-$Ca^{2+}$
(not shown) suggests that very little of the sulphate is present as $CaSO_4$, either formed by heterogeneous
deposition of $SO_2$ on calcite mineral particles or liberated from the soils as mineral gypsum (Annegarn et
al., 1983).
Finally, the mean annual concentration of oxalate at HBAO was 72 ± 80 ng m$^{-3}$ in 2016 and 141 ± 50 ng m$^{-3}$
in 2017. Values at HBAO are consistent with those reported by Zhang et al. (2010) over the southeast
Atlantic (200 ± 140 ng m$^{-3}$). Oxalate aerosols in the atmosphere are due to marine biogenic activity and





anthropogenic emissions including fossil fuel combustion and biomass burning (Gillett et al., 2007, and
references therein). It is also formed by in-cloud processes and oxidation of gaseous precursors followed
by condensation (Baboukas et al., 2000). The moderate correlation with $NO_3^-$, $nss\text{-}SO_4^{2-}$ and $nss\text{-}K^+$,
particularly in 2017, could suggest a common origin and possible influence of occasional biomass burning.
### 4.3    Source apportionment
Figure 8 presents a summary of the five source profiles resolved by the selected PMF solution. According
to their characteristic tracers, the five sources are labelled sea salt ($Na^+$, $Cl^-$, $F^-$, $Mg^{2+}$, $ss\text{-}SO_4^{2-}$, $ss\text{-}K^+$ and $ss\text{-}$
$Ca^{2+}$), marine biogenic (MSA, oxalate and As), mineral dust (Si, Al, Fe, Ti, $F^-$, $nss\text{-}K^+$ and $nss\text{-}Ca^{2+}$), secondary
products ($nss\text{-}SO_4^{2-}$, $NH_4^+$, formate, oxalate and $nss\text{-}K^+$) and heavy metals (V, Ni, Zn, Pb, Cr, Cu, Cd, Nd, Sr
and Co). For the sake of comparison, the mass concentration of sea salt and mineral dust can also be
evaluated from the chemical composition as:
[Sea salt] = 2.57 *[$Na^+$], as in Zhang et al. (2010)    (1a)
[MD] = [Mineral dust] = [Al]/0.0813 + 2.5 * [$nss\text{-}Ca^{2+}$], as explained in section 3.1    (1b)
The estimated aerosol mass (EAM) needed to evaluate their percent contributions can then be calculated
as:
[EAM] = [Sea salt] + [Mineral dust] + [other]    (1c)
Where [other] represents the major compounds (by mass) in the input dataset, as:
[other] = [V]+[Ni] + [Zn] + [Pb] + [Cr] + [Cu] + [$nss\text{-}SO_4^{2-}$] + [$NH_4^+$] + [MSA] + [$NO_3^-$] + [Oxalate]    (1d)
#### 4.3.1    Sea salt
Due to the location of the sampling site in this windy, coastal region, sea salts were present to some extent
in all the PMF resolved sources. The sea salt source profile is described by typical elements like sodium,
chloride, fluoride and magnesium ions as well as the sea salt fraction of potassium, sulphate and calcium
(Figure 8). These species accounted for 70.8 ± 0.2% of the total estimated mass concentration, in good
agreement with the percent contribution of the sea salt mass from Equation 1a (70.4%). In general terms,
the source profile obtained by PMF is in good agreement with that obtained by chemical apportionment. In
particular, the $Cl^-/Na^+$ mass ratio is 1.4 ± 1.6, confirming the depletion of $Cl^-$ with respect to $Na^+$. The mass
ratio of 0.11 ± 0.01 for $Mg^{2+}/Na^+$ estimated in this source was in good agreement to that reported in Table
2 and for the average seawater composition. The $F^-/Na^+$ mass ratio reported in Table 2 is higher than 0.19
± 0.12 for this PMF sea salt source and this ratio is still two orders of magnitude higher than the average
seawater composition reported in Seinfeld and Pandis (2006).





A separate PMF analysis was run with $Ca^{2+}$, $SO_4^{2-}$ and $K^+$ ions without separating the sea salt from non-sea
salt components. The contributions of these elements to the sea salt aerosol mass were estimated to be 55
$\pm$ 0.1% of the calcium, 60.1 $\pm$ 0.1% of the sulphate and 72.9 $\pm$ 0.1% of the potassium. This compared well to
the PMF solution of the dataset with the species separated by ss and nss components, indicating good
agreement in the apportionment by nominal mass ratios and PMF of the sea salt source.

### 519    4.3.2    Marine biogenic

The marine biogenic source profile contributed 13.5 $\pm$ 0.8% to the $PM_{10}$ mass concentration. The source
profile is composed of secondary products emitted by marine biogenic processes such as MSA, oxalate and
As (Figure 8), which however only account for 2.8% ($\pm$ 0.1%) of the total mass. The largest fraction of the
mass is contributed by $Cl^-$ and $Na^+$ (combined mass contribution of 67.3 $\pm$ 0.1%) and ss- and nss-sulphates
(12.7 $\pm$ 0.1%). The MSA/nss-$SO_4^{2-}$ mass ratio for this source is 0.05 $\pm$ 0.25 and compares well to the chemical
apportionment and what was measured in the region by Andreae et al. (1995).
The fact that we find arsenic in this source is interesting, as it was not immediately evident from the analysis
of correlations between the two, but it does confirm the contributions from marine biogenic sources
(Sanders and Windom, 1980; Shibata et al., 1996).

### 529    4.3.3    Mineral dust

The contribution of mineral dust to the $PM_{10}$ aerosol mass estimated by the PMF analysis is 9.9% ($\pm$ 0.1%),
lower than the average of 17% calculated with Equation 1b. The difference in the estimated aerosol mass
is then likely explained by the use of the average aluminium content in global soils (8.13%) in the
calculation of estimated aerosol mass, indicating a significant difference in aluminium content of Namibian
soil as compared to the global average.
The mineral dust source profile was composed by Si, Al, Fe, Ti, $F^-$, nss-$K^+$ and nss-$Ca^{2+}$ (Figure 8), accounting
for almost all of its evaluated mass concentration (94.9 $\pm$ 0.1%). The mass contributions of P, Mn and V to
the source profile was small (0.3$\pm$ 0.1%). Nevertheless, mineral dust accounts for up to 40 to 55% of their
variance in mass. The vanadium in this source was highly enriched to the nickel in the V/Ni mass ratio of
32.74 $\pm$ 0.01, not consistent with shipping emissions reported for the Mediterranean (Becagli et al., 2017;
Viana et al., 2009).
The nss-$K^+$/Al mass ratio was estimated as 0.07 $\pm$ 0.10, lower than that obtained by chemical
apportionment (0.13 $\pm$ 0.12) and those reported in the literature (0.25 – 0.45, Eltayeb et al., 1993). The PMF
allocated 71.4 $\pm$ 0.1% of the nss-$K^+$ to this source and the remaining 28.4 $\pm$ 0.1 % to the marine biogenic
and secondary products sources.



The nss-$SO_4^{2-}$/nss-$Ca^{2+}$ mass ratio was found to be 0.25 in both the PMF resolved source and the chemical
apportionment and was an order of magnitude lower than the mass ratio for gypsum indicating that
gypsum was not a main sulphate component in the region. The PMF attributed 35.8 ± 0.1% of the total $Ca^{2+}$
mass to this source, slightly lower than that calculated by nominal mass ratios for the nss-$Ca^{2+}$(45%). Some
very high concentrations in fluoride were associated with calcium and indicate a likely source of calcium
fluoride minerals such as fluorspar, presently mined at Okorusu mine.

### 4.3.4   Secondary products

The secondary products source profile, with an estimated $PM_{10}$ aerosol mass of 3.2 ± 1.0%, was identified
by secondary nss-$SO_4^{2-}$, $NH_4^+$ and formate (Figure 8). These species account for 61.6 ± 0.3% of the source
mass. MSA, nitrate, oxalate and nss-$K^+$ contribute between 15 and 30% of their masses to this source and
together account for an additional 17.2 ± 0.1% of the source mass.
The $NH_4^+$/nss-$SO_4^{2-}$ mass ratio for this source was 0.23 ± 0.33, consistent with ammonium bisulphate.
Sources of nss-$SO_4^{2-}$ and $NH_4^+$ include gaseous precursors released from various biogenic and
anthropogenic sources, including biomass burning (Andreae et al., 1995; Behera et al., 2013; Theobald et
al., 2006; Zhang et al., 2010). The fact that formate, released by the oxidation of DMS and also during
biomass burning (Finlayson-Pitts and Pitts Jr, 2000, and references therein) was contributed to this source
was not surprising as it is only stable in high concentrations of ammonia (Andreae, 2000). The
corresponding contribution of nitrate, oxalate and nss-$K^+$ to this source might suggest a biomass burning
source (Andreae et al., 1998; Andreae and Merlet, 2001; Gillett et al., 2007), but the oxalate/nss-$SO_4^{2-}$ and
nss-$K^+$/nss-$SO_4^{2-}$ for this source were lower than the values reported for biomass burning emissions in
Formenti et al. (2003b) and Andreae et al. (1998).
The MSA/nss-$SO_4^{2-}$ mass ratio for this source was 0.03 ± 0.02, inconsistent with other literature on marine
biogenics reported in Table 4, as opposed to the marine biogenic source identified by PMF which is in good
agreement with the ratio reported in Andreae et al. (1995).

### 4.3.5   Heavy metals

The fifth and final source profile (2.3 ± 2.5% to the total mass), was characterised by the heavy metals V,
Ni, Zn, Pb, Cr, Cu, Cd, Nd, Sr and Co which together contribute to 34.1 ± 0.8% of the source mass. Sea salt
species contributed 68.4 ± 0.2% of the mass of this source. Some of these metals were associated with
mineral dust and the additional portions of these metals may be contributed by commercial shipping or
industrial processes, as discussed previously. The V/Ni ratio for this source was 1.00 ± 1.00, lower than the
average ratios of 1.7 ± 1.1 in 2016 and 1.3 ± 1.3 in 2017 from the chemical apportionment and lower than



2.8 – 5 reported for different locations across the Mediterranean (Becagli et al., 2017; Schembari et al.,
2014; Viana et al., 2009).
Small amounts of $nss\text{-}SO_4^{2-}$ were contributed to this source and the $nss\text{-}SO_4^{2-}/V$ mass ratio was $23.5 \pm 1.75$,
in good agreement with 16 – 24 reported by Alföldy et al., (2013) for Rotterdam harbour. Although some
of these heavy metals may be sourced from the commercial shipping route offshore, the mass ratios for
tracer elements were not in agreement with our results and so we cannot conclusively state shipping fuel
combustion as the source of these heavy metals.
**5.  Conclusions and significance of results**
This paper presented the first long-term characterisation of the elemental and ionic composition of
atmospheric aerosols and the source apportionment of the $PM_{10}$ mass fraction at the Henties Bay Aerosol
Observatory on the west coast of southern Africa, an under-explored region of the world to date.
The study was based on semi-continuous filter sampling at the HBAO site in Namibia in 2016 and 2017,
laboratory analysis of the collected samples by X-ray fluorescence and ion chromatography, and PMF
apportionment, supported by back-trajectory calculations and the analysis of local winds.
Trajectory analysis for the sampling period from 2016 to 2017 shows four distinct patterns of atmospheric
transport to HBAO. Two transport pathways are from the South Atlantic Ocean, namely directly from the
east and the south and south-east. A third transport pathway shows air masses reaching Henties bay from
the north-west.  This pathway will likely include constituents that originated over the continent. The final
transport pathway is from central southern Africa.
Local wind circulation is influenced by the overlying synoptic circulation patterns as well as local sea breeze
mechanisms. Surface flow to HBAO is predominantly from the south and south-west. Southwesterly flow is
likely to be linked to sea breeze circulation as a result of thermal gradients in the daytime between the
desert surface and the ocean. Land and sea breezes are not common at HBAO due to a weak thermal
gradient at night between the ocean and desert surface.
In general terms, the results presented in this paper are in fair agreement with the expectations for remote
marine regions of the world, and previous observations in the area (Andreae et al., 1995; Zhang et al., 2010).
The $PM_{10}$ aerosol load is dominated by natural species such as sea salt, mineral dust and marine biogenic
emissions, accounting for more than 90% of the mass. As a consequence of the proximity to the seashore of
the HBAO sampling station, the majority of the $PM_{10}$ mass concentration (70%) is due to sea spray, which
is persistent at the diurnal and seasonal time scales.




For the first time, the frequency, intensity, and elemental composition of Namibian mineral dust aerosols
were investigated. Nineteen episodes of increased Al and nss-Ca$^{2+}$ concentrations and lasting from one to a
maximum of four days were detected during the entire sampling period. This corresponds well to the
frequency of emission of dust plumes from river valleys, coastal sabkhas, and paleo-lacustrine sources
(Etosha and Makgadikgadi pans) observed by various authors (Eckardt and Kuring, 2005; Vickery et al.,
2013; Dansie et al., 2017). Our data series does not show any particular time dependence of the frequency
nor duration of the detected episodes. This is in contrast with the observation by Dansie et al. (2017) that
windblown dust derived from the ephemeral river valleys is transported offshore during large easterly
wind events, and indicative of the fact that HBAO is the receptor of mineral dust emitted by various sources.
One of the striking findings of this paper was the level of anthropogenic contamination and the
concentrations of various pollutants, including heavy metals and fluoride. Formenti et al. (2018) already
demonstrated a seasonal increase in the light-absorbing carbon particulate between May and late July,
indicative of the surface transport of biomass burning aerosols, and episodically throughout the year,
attributed to pollution by ship traffic along the Cape of Good Hope sea route. This work additionally
demonstrates that mining activities severely affect the air quality and contribute to concentrations as high
as, or even higher than in well-known polluted regions of the world, such as the Venice lagoon (Prodi et al.,
2009). The persistence of these high concentrations over the two years of sampling is extremely worrying
for the affected populations and needs to be addressed by dedicated investigations and decision-making
procedures. We suspect that some of that contamination, contributing to the highest heavy metal
concentrations in October 2016, might be due to the major road construction between Walvis Bay, past
Henties Bay and towards Angola, that started in the second half of 2016. Having said this, despite maximum
concentrations being measured at that time, there is no significant difference between the concentration
levels in 2016 (before road works) and 2017 (during the road works), suggesting that the pollution by
heavy metals is a specific feature in the region, which likely has implications on weather and climate. One
such effect could be the deposition of these metals in the ocean. While the deposition of nutrients from the
outflow of mineral dust could be important in fertilising waters near the coast (Dansie et al., 2017) and in
the Southern Ocean (Okin et al., 2011), the deposition of the trace metals (Cr, Cu, Ni, Mn, or Zn) in aerosols,
which play a biological role in enzymes and as structural elements in proteins (Morel and Price, 2003),
could also affect the marine productivity of the BUS, one of the most productive marine environments in
the world, and should be explored. The complexity and diversity of sources that might contribute to the
mineral dust load at HBAO over a year, as well as the detailed chemical composition including trace metal
contamination, deserve certainly further dedicated investigation.



The long-term time series of aerosol composition at HBAO also provides new and important insights on the
contribution of marine emission to the regional aerosol load. Our sampling provides the first long-term
measurements of the mass concentrations of MSA in the south Atlantic, and the apportionment of sulphate
aerosols, which are important for light scattering and cloud formation. Our data show that sea salt
contributes, on average, to around 57% of the total sulphate mass. The non-sea salt fraction (nss-$SO_4^{2-}$), of
the order of 43%, is partly attributed to the oxidation of sulphur-containing gaseous emissions (DMS, $SO_2$,
$H_2S$) during the marine phytoplankton life cycle, likely favoured by night-time fog and overall elevated
relative humidity, typical along the coast. However, nss-$SO_4^{2-}$ mass concentrations over 2 µg m$^{-3}$ could be
contributed by ship plumes as well as by episodic biomass burning. Ammonium bisulfate (($NH_4$)$HSO_4$) was
found to be the predominant sulphate forms at HBAO, where, incidentally, we observed dramatic rusting
and corrosion of materials through the years. The ongoing data analysis of the AEROCLO-sA field campaign
will provide with further insights on the size-dependent apportionment, chemical composition and
hygroscopicity of sulphate aerosols.
***Data availability.*** Original and analysed data can be obtained by email request to the corresponding author. The
SplitR package is found in Iannone (2018). The openair package for R is found in Carslaw and Ropkins (2017). The
EPA (Environmental Protection Agency) PMF version 5.0 software is available from https://www.epa.gov/air-
research/positive-matrix-factorization-model-environmental-data-analyses. The NOAA Air Resources Laboratory
(ARL) provides the HYSPLIT transport and dispersion model and/or READY website (http://www.ready.noaa.gov).
***Author contributions.*** DK, PF, SJP, AN, MC, and AF performed the filter sampling and operated the wind sensor. PH,
SC, FL, CMB, ST, and ZZ performed the XRF and IC analysis of the collected samples. DK performed the back-trajectory
calculations, analysis of wind data and PMF. DK and PF performed the analysis of the chemical analysis and integration
of the dataset. DK and PF wrote the paper with contributions from SJP.
***Competing interests.*** The authors declare that they have no conflict of interest.
***Special issue statement***. This article is part of the special issue "New observations and related modelling studies of
the aerosol–cloud–climate system in the Southeast Atlantic and southern Africa regions (ACP/AMT inter-journal SI)".
It is not associated with a conference.
**Acknowledgements.** This work receives funding by the French Centre National de la Recherche Scientifique (CNRS)
and the South African National Research Foundation (NRF) through the "Groupement de Recherche Internationale
Atmospheric Research in southern Africa and the Indian Ocean" (GDRI-ARSAIO) and the Project International de
Coopération Scientifique (PICS) "Long-term observations of aerosol properties in Southern Africa" (contract n.
260888) as well as by the Partenariats Hubert Curien (PHC) PROTEA of the French Minister of Foreign Affairs and
International Development (contract numbers 33913SF and 38255ZE). D. Klopper acknowledges the financial



support of the Climatology Research Group of North-West University and the travel scholarship of the French Embassy
in South Africa.



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





**Table captions**
**Table 1**. Summary statistics of elemental and water-soluble ionic concentrations measured at HBAO. The
second column indicates the number of samples for which values were above the minimum quantification
limit (MQL). The arithmetic means with standard deviations (sd) and range of mass concentrations
(minimum and maximum) are given in ng m$^{-3}$.
**Table 2**. Annual mean mass ratios of Cl$^-$, Mg$^{2+}$, K$^+$, Ca$^{2+}$, F$^-$ and SO$_4^{2-}$ with respect to Na$^+$ for 2016 and 2017
as well as the $R^2$ for each linear model and mass ratios for seawater reported in the literature (Seinfeld and
Pandis; 2006). Standard deviations are indicated as *sd*.
**Table 3.** Average mass ratios and standard deviations for 2016 and 2017 for dust episodes. Mass ratios
reported in the literature are given in the last row for comparison.
**Table 4.** Reported concentrations for marine biogenic and secondary aerosols for different locations, and
especially in the southern hemisphere.  Concentrations are in μg m$^{-3}$ unless stated otherwise.
**Figure captions**
**Figure 1.** Geographical map of Namibia with elevation as a shaded gradient and some of the known
emission sources in the region, such as major urban settlements and airports, harbours, pans and swamps,
mineral-rich mining operations, labelled by the major element begin mined, and dune fields of the Kalahari
stratigraphic group (Atlas of Namibia project, 2002).
**Figure 2.** Composite maps of 72-hour back-trajectories for every filter sampling period in 2016 (dates in
blue) and 2017 (dates in orange). From these composite maps, a clear distinction can be made between
marine air masses and those of continental origin and the potential for variability from these regions in
terms of distance travelled and trajectory pathway. The colours are only used to differentiate one set of
trajectories from another.
**Figure 3.** Wind roses showing the wind speed, direction and frequency of occurrence corresponding to
each aerosol sampling week in 2016 (dates in blue), and 2017 (dates in orange). The arithmetic mean wind
speed for each week is reported in green. For 7–14 July 2017 no surface wind data is available.
**Figure 4.** Hourly wind roses during the aerosol sampling at HBAO. The arithmetic means and percentage
of calm conditions, when wind speeds are below detection, is reported in green. Time is in UTC. For 7–14
July 2017 no surface wind data is available.





**Figure 5.** Time series (datetime in UTC) of measured concentrations for $Na^+$, $Ca^{2+}$, Al, $K^+$, $SO_4^{2-}$, MSA and Ni (shaded area). The solid black line indicates the calculated 10-point moving average. The sea salt (ss) components for $Ca^{2+}$, $K^+$ and $SO_4^{2-}$ is indicated by the orange shaded areas, the non-sea salt (nss) fraction is represented by the blue shaded areas. The time series is non-consecutive and is divided into the 26 sampling weeks by the light grey vertical lines.

**Figure 6.** Scatterplots of nss-$Ca^{2+}$/Al (top left), nss-$K^+$/Al (top right), V (bottom left) and Ni (bottom right) ratios to Al for 2016 (blue) and 2017 (orange). Concentrations are expressed in $\mu g\ m^{-3}$. Note the logarithmic y-axes on the top plots.

**Figure 7.** Scatterplots for ratios of MSA (left) and $NH4+$ (right) to nss-$SO42-$ for 2016 (blue) and 2017 (orange). Concentrations are expressed in $\mu g\ m-3$. Note the logarithmic y-axis of the figure on the right.

**Figure 8.** Profiles of the five sources identified by the PMF analysis. Blue bars denote the mass concentrations of individual elements/ionic species (left logarithmic axis, $ng\ m^{-3}$) while the yellow points indicate the percent of species attributed to the source (right axis).





**Table 1.** Summary statistics of elemental and water-soluble ionic concentrations measured at HBAO. The second
column indicates the number of samples for which values were above the minimum quantification limit (MQL). The
arithmetic means with standard deviations (sd) and range of mass concentrations (minimum and maximum) are given
in ng m$^{-3}$.

| Chemical species | Number of samples | Mean ± sd | Range |
|---|---|---|---|
| Cl | 385 | 13216 ± 7987 | 17 - 50041 |
| S | 383 | 1346 ± 645 | 1 - 4386 |
| Ca | 366 | 885 ± 768 | 75 - 6862 |
| Fe | 383 | 367 ± 458 | 3 - 3687 |
| Na | 380 | 8435 ± 5752 | 18 - 42688 |
| Mg | 380 | 1178 ± 792 | 1 - 6416 |
| Al | 379 | 478 ± 581 | 2 - 4739 |
| Si | 374 | 1687 ± 2102 | 5 - 17016 |
| P | 352 | 10 ± 8 | 1 - 72 |
| K | 379 | 511 ± 359 | 8 - 3076 |
| Ti | 367 | 39 ± 47 | 1 - 363 |
| Mn | 295 | 13 ± 11 | 1 - 86 |
| Zn | 182 | 12 ± 7 | 1 - 42 |
| Cr | 228 | 8 ± 6 | 1 - 31 |
| V | 334 | 8 ± 5 | 1 - 38 |
| Ba | 100 | 9 ± 7 | 1 - 34 |
| Co | 261 | 8 ± 5 | 1 - 32 |
| Cu | 228 | 13 ± 9 | 1 - 48 |
| Nd | 296 | 15 ± 11 | 1 - 61 |
| Ni | 278 | 8 ± 6 | 1 - 33 |
| Sr | 251 | 77 ± 63 | 2 - 346 |
| Cd | 214 | 735 ± 1124 | 1 - 6776 |
| As | 221 | 191 ± 317 | 1 - 1092 |
| Pb | 193 | 75 ± 89 | 1 - 509 |
| F$^-$ | 375 | 3356 ± 3201 | 110 - 25240 |
| Acetate | 90 | 27 ± 36 | 11 - 235 |
| Propionate | 79 | 46 ± 21 | 12 - 162 |
| Formate | 322 | 23 ± 12 | 5 - 73 |
| MSA | 330 | 63 ± 38 | 11 - 232 |
| Cl$^-$ | 376 | 13980 ± 9834 | 117 - 76008 |
| Br$^-$ | 17 | 44 ± 15 | 27 - 77 |
| NO$_3^-$ | 364 | 232 ± 432 | 26 - 8167 |
| PO$_4^-$ | 41 | 60 ± 62 | 27 - 397 |
| SO$_4^{2-}$ | 376 | 3602 ± 1853 | 81 - 14331 |
| Oxalate | 379 | 121 ± 53 | 13 - 474 |
| Na$^+$ | 376 | 10199 ± 6853 | 32 - 52987 |
| NH$_4^+$ | 376 | 205 ± 126 | 25 - 1747 |
| K$^+$ | 373 | 413 ± 265 | 23 - 1976 |
| Mn$^{2+}$ | 7 | 41 ± 35 | 22 - 117 |
| Ca$^{2+}$ | 371 | 727 ± 618 | 35 - 5232 |
| Mg$^{2+}$ | 370 | 1168 ± 768 | 29 - 5585 |






**Table 2.** Annual mean mass ratios of Cl⁻, Mg²⁺, K⁺, Ca²⁺, F⁻ and SO₄²⁻ with respect to Na⁺ for 2016 and 2017 as well as
the $R^2$ for each linear model and mass ratios for seawater reported in the literature (Seinfeld and Pandis; 2006).
Standard deviations are indicated as *sd*.

|  | 2016 | | 2017 | | Average seawater |
|---|---|---|---|---|---|
|  | Mass ratio ± sd | $R^2$ | Mass ratio ± sd | $R^2$ | Mass ratio |
| Cl⁻/Na⁺ | 1.35 ± 0.11 | 0.99 | 1.34 ± 0.11 | 0.99 | 1.80 |
| Mg²⁺/Na⁺ | 0.12 ± 0.01 | 0.99 | 0.11 ± 0.01 | 0.99 | 0.12 |
| K⁺/Na⁺ | 0.04 ± 0.01 | 0.98 | 0.04 ± 0.01 | 0.93 | 0.04 |
| Ca²⁺/Na⁺ | 0.07 ± 0.04 | 0.61 | 0.07 ± 0.05 | 0.42 | 0.04 |
| SO₄²⁻/Na⁺ | 0.36 ± 0.14 | 0.95 | 0.42 ± 0.23 | 0.85 | 0.25 |
| F⁻/ Na⁺ | 0.38 ± 0.24 | 0.53 | 0.32 ± 0.35 | 0.33 | 0.000122 |







Table 3. Average mass ratios and standard deviations for 2016 and 2017 for dust episodes. Mass ratios reported in
the literature are given in the last row for comparison.

| | 2016 | | 2017 | | Dust episodes | | Literature values |
|---|---|---|---|---|---|---|---|
| | Mean (± sd) | $R^2$ | Mean (± sd) | $R^2$ | Mean (± sd) | $R^2$ | |
| Si/Al | 3.75 ± 1.04 | 0.96 | 3.51 ± 0.74 | 0.96 | 3.60 ± 0.65 | 0.94 | 2.87 – 6.13[a], 3.41[b], 4.63[c] |
| nss-Ca$^{2+}$/Al | 1.27 ± 0.67 | 0.89 | 0.5 ± 0.37 | 0.83 | 1.42 ± 0.87 | 0.60 | 0.35 – 6.06[a], 0.19[c] |
| Fe/Al | 0.75 ± 0.19 | 0.96 | 0.81 ± 0.20 | 0.97 | 0.81 ± 0.11 | 0.97 | 0.65 – 1.06[b], 0.53[c] |
| V/Al | 0.03 ± 0.03 | 0.37 | 0.03 ± 0.04 | 0.26 | 0.02 ± 0.03 | 0.31 | 0.0014[c] |
| Ti/Al | 0.07 ± 0.22 | 0.96 | 0.1 ± 0.04 | 0.97 | 0.08 ± 0.02 | 0.97 | 0.09 – 0.15[a], 0.07[c] |
| P/Al | 0.03 ± 0.02 | 0.81 | 0.03 ± 0.03 | 0.59 | 0.02 ± 0.01 | 0.72 | 0.007[d] |
| Fe/nss-Ca$^2$ | 0.66 ± 0.23 | 0.94 | 2.73 ± 25.81 | 0.83 | 8.57 ± 77.71 | 0.60 | 0.18 – 1.86[a], 0.58[b], 2.77[c] |
| nss-K$^+$/Al | 0.13 ± 0.12 | 0.81 | 0.13 ± 0.13 | 0.59 | 0.08 ± 0.06 | 0.61 | 0.251 – 0.452[a] |
| V/Si | 0.01 ± 0.01 | 0.39 | 0.01 ± 0.01 | 0.26 | 0.01 ± 0.01 | 0.33 | 0.0003[c] |
| F$^-$/Al | 11.75 ± 8.40 | 0.73 | 9.66 ± 8.38 | 0.64 | 6.90 ± 4.56 | 0.57 | – |
| nss-SO$_4^{2-}$/nss-Ca$^{2+}$ | 5.32 ± 6.79 | 0.08 | 8.72 ± 44.24 | 0.03 | 3.21 ± 5.73 | 0.11 | 2.4[e] |

[a] Eltayeb et al., (1993) from various sites around the Central Namib
[b] Annegarn et al. (1983): Gobabeb, Namibia
[c] Seinfeld and Pandis (2006): average chemical composition for soils globally
[d] Formenti et al., (2003): Cape Verde region
[e] Molar mass ratio for gypsum





**Table 4.** Reported concentrations for marine biogenic and secondary aerosols for different locations, and especially
in the southern hemisphere.  Concentrations are in μg m⁻³ unless stated otherwise.

|  | $SO_4^{2-}$ (nss-$SO_4^{2-}$) | $NH_4^+$ | $NO_3^-$ | MSA | MSA/nss-$SO_4^{2-}$ |
|---|---|---|---|---|---|
| **Outflow Africa south of Cape Town, PM1** [a] | 1.39 | 0.18 | 0.01 | 0.04 | 0.007 [aaa] |
| **Southern Ocean south of Australia** [b] | - | - | - | 0.02 - 0.2 | 0.24 ± 0.16 |
| **Cape Grim, Tasmania** [c] | 11.9 ± 1.2 nmole/m³ | - | - | 0.167 ± 0.027 nmole/m³ | 0.063 ± 0.020 |
| **19°S offshore southern Africa** [d] | - | - | - | 6.1 ± 4 ppt; 6.3 ± 4.4 ppt | 0.05 - 0.11 |
| **Southern Atlantic** [e] **A = autumn, S = spring** [f] | 1.95 ± 1.05 [e] | 7.6 ± 13.9 [e] | 1.05 ± 0.72 [e] | 0.21 ± 0.30 [e] S: 0.05 ± 0.1 [j] A: 0.15 ± 0.1 [j] | 0.11 [e] |
| **Southern Pacific** [e] | 2.10 ± 1.05 | 0 ± 0 | 0.12 ± 0.15 | 0.58 ± 0.60 | 0.27 |
| **Venice Lagoon** [g] **W = winter, S = spring** | W: 3.3 ± 1.0; S: 4.4 ± 1.2 | W: 2.9 ± 0.6 S: 2.6 ± 1.0 | W: 9.0 ± 2.4 S: 3.5 ± 2.9 | W: 0.035 ± 0.017 S: 0.054 ± 0.040 | 0.1 |
| **Southern Indian Ocean** [h] | - | - | - | - | 0.1 |
| **America Samoa (14°S, 170°W)** [i] | - | - | - | - | 0.06 |
| **Coastal Antarctica** [j] | - | - | - | - | 0.05 - 0.17 |
| **This study; 2016** | 4.0 ± 2.4 (1.7 ± 0.8) | 0.19 ± 0.10 | 0.26 ± 0.71 | 0.07 ± 0.01 | 0.03 ± 0.01 |
| **This study; 2017** | 3.4 ± 1.4 (1.6 ± 0.7) | 0.20 ± 0.10 | 0.22 ± 0.12 | 0.07 ± 0.04 | 0.04 ± 0.02 |

[a] Zorn et al. (2008); PM1 fraction, [aaa] calculated with respect to total sulphate
[b] Quinn et al. (1998)
[c] Ayers et al. (1986)
[d] Andreae et al. (1995)
[e] Zhang et al. (2010); total suspended particulates fraction
[f] Huang et al. (2016)
[g] Prodi et al. (2009)
[h] Sciare et al. (2000)
[i] Savoie et al. (1994)
[j] Chen et al. (2012)





**Figure 1.** *Geographical map of Namibia with elevation as a shaded gradient and some of the known emission sources in the region, such as major urban settlements and airports, harbours, pans and swamps, mineral-rich mining operations, labelled by the major element begin mined, and dune fields of the Kalahari stratigraphic group (Atlas of Namibia project, 2002).*

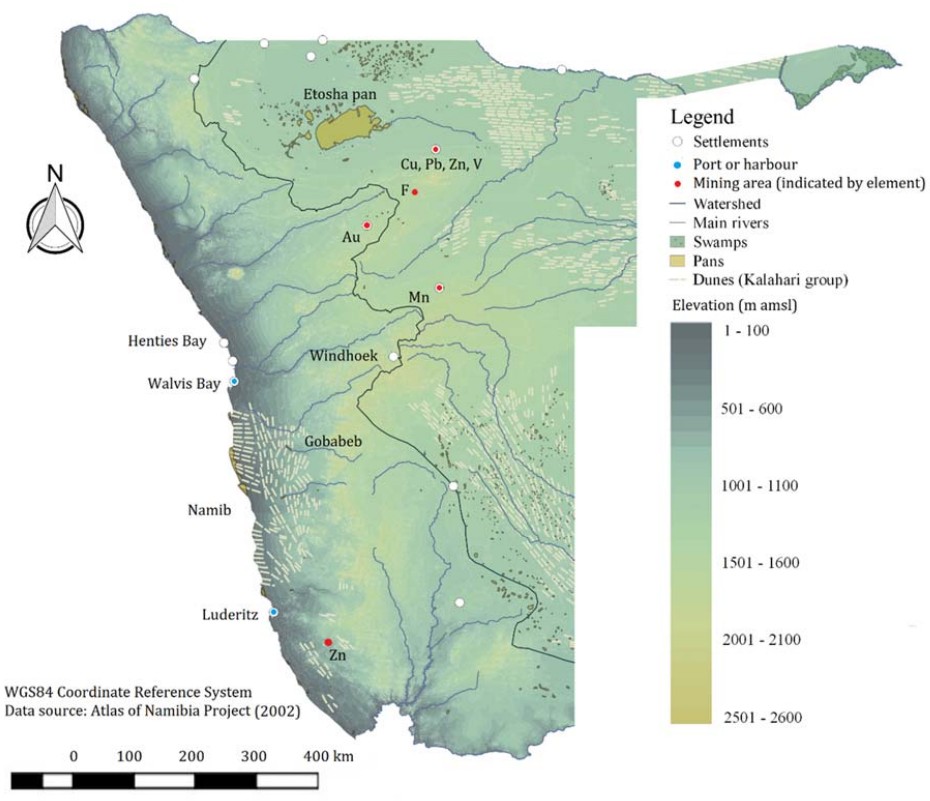





**Figure 2.** *Composite maps of 72-hour back-trajectories for every filter sampling period in 2016 (dates in blue) and 2017*
*(dates in orange). From these composite maps, a clear distinction can be made between marine air masses and those of*
*continental origin and the potential for variability from these regions in terms of distance travelled and trajectory*
*pathway. The colours are only used to differentiate one set of trajectories from another.*

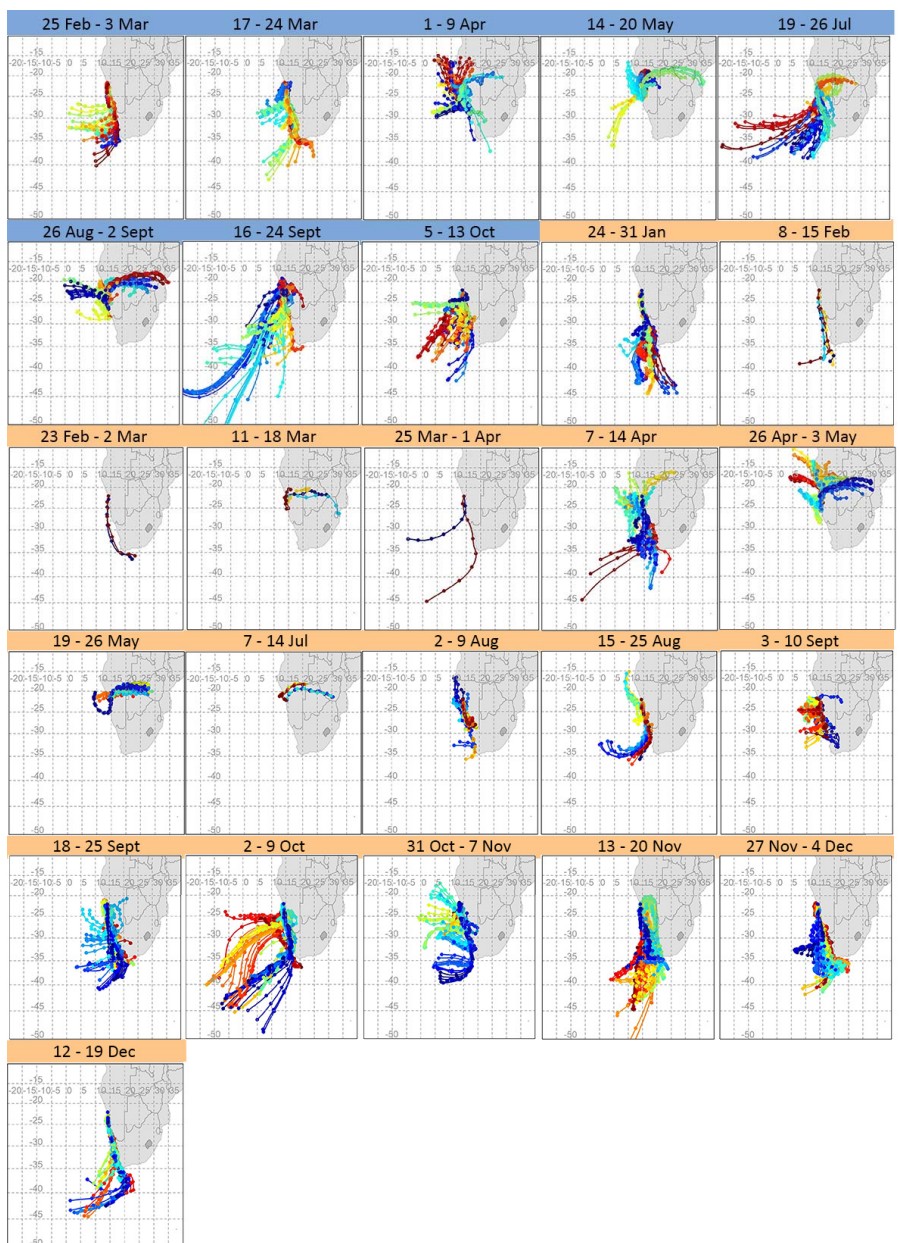



*Figure 3. Wind roses showing the wind speed, direction and frequency of occurrence corresponding to each aerosol*
*sampling week in 2016 (dates in blue), and 2017 (dates in orange). The arithmetic mean wind speed for each week is*
*reported in green. For 7–14 July 2017 no surface wind data is available.*

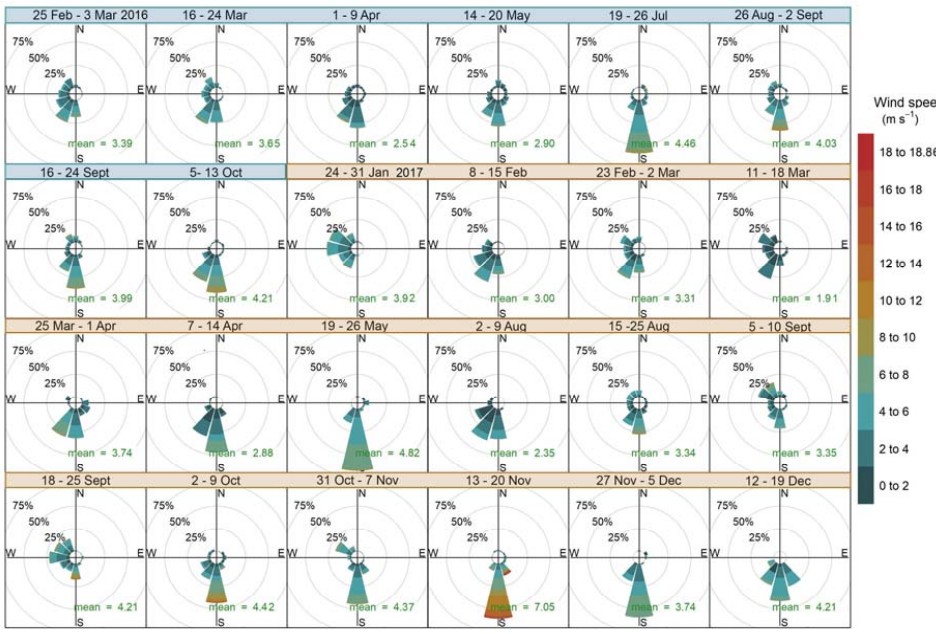





**Figure 4.** *Hourly wind roses during the aerosol sampling at HBAO. The arithmetic means and percentage of calm*
*conditions, when wind speeds are below detection, is reported in green. Time is in UTC. For 7–14 July 2017 no surface*
*wind data is available.*

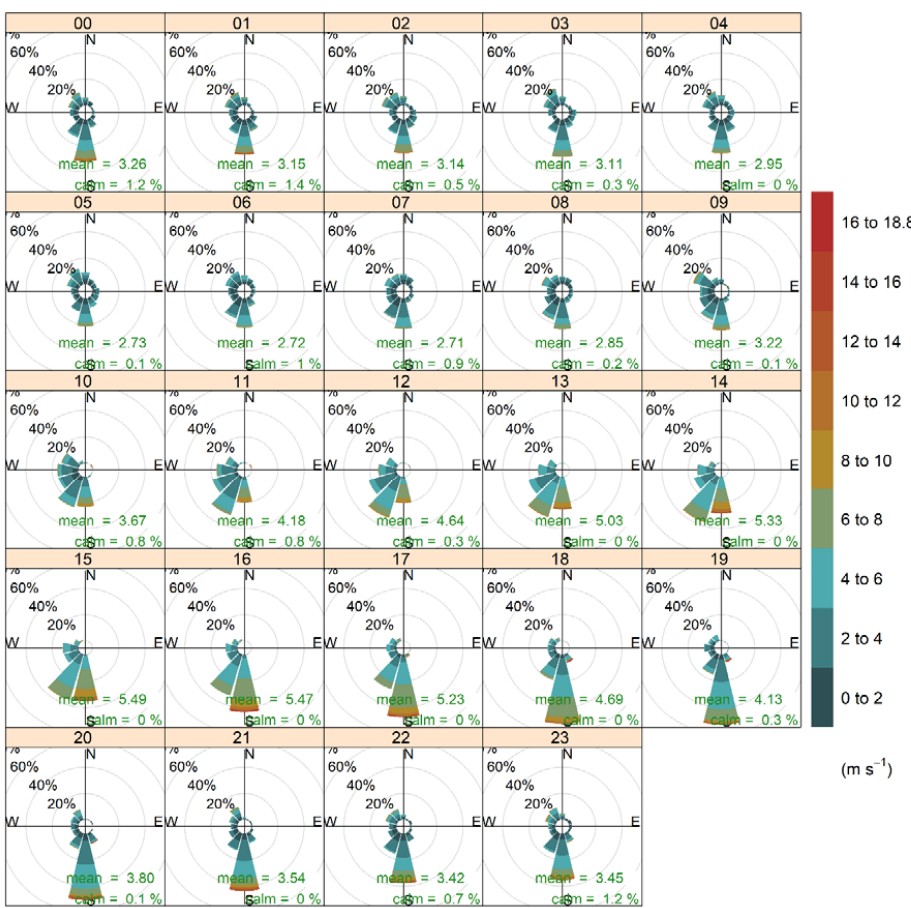

**Frequency of counts by wind direction (%)**







**Figure 5.** *Time series (datetime in UTC) of measured concentrations for Na+, Ca2+, Al, K+, SO42-, MSA and Ni (shaded area).*
*The solid black line indicates the calculated 10-point moving average. The sea salt (ss) components for Ca2+, K+ and SO42-*
*is indicated by the orange shaded areas, the non-sea salt (nss) fraction is represented by the blue shaded areas. The time*
*series is non-consecutive and is divided into the 26 sampling weeks by the light grey vertical lines.*

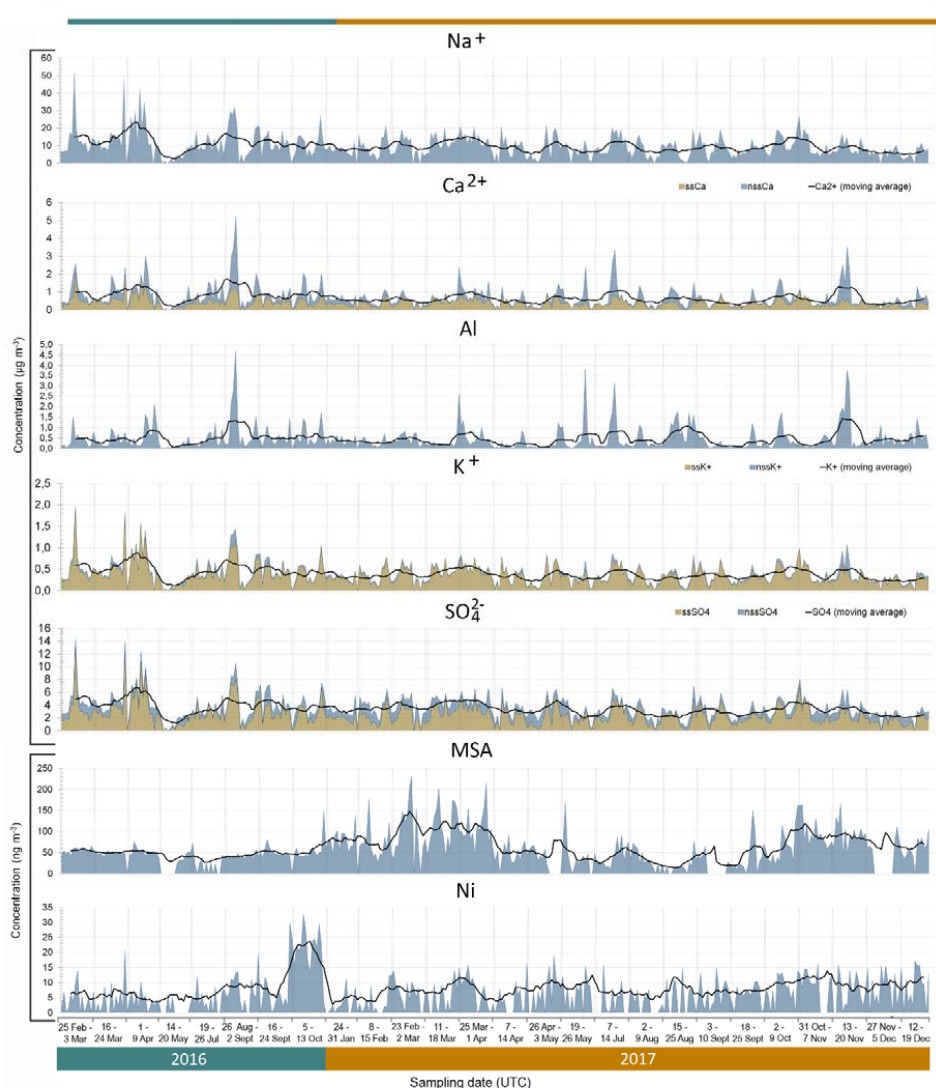







**Figure 6.** *Scatterplots of nss-Ca²⁺/Al (top left), nss-K⁺/Al (top right), V (bottom left) and Ni (bottom right) ratios to Al for*
*2016 (blue) and 2017 (orange). Concentrations are expressed in μg m⁻³. Note the logarithmic y-axes on the top plots.*

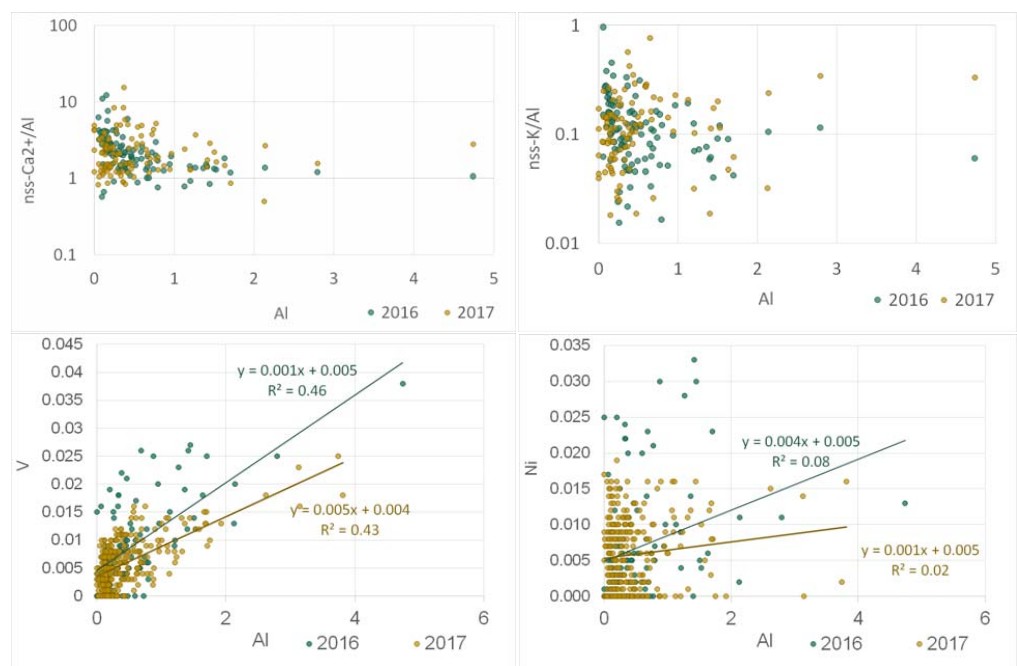








**Figure 7.** *Scatterplots for ratios of MSA (left) and NH4+ (right) to nss-SO42- for 2016 (blue) and 2017 (orange).*
*Concentrations are expressed in μg m-3. Note the logarithmic y-axis of the figure on the right.*

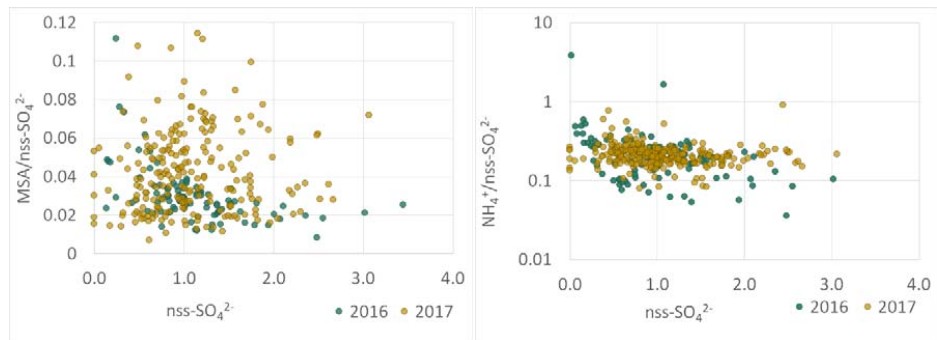





**Figure 8.** *Profiles of the five sources identified by the PMF analysis. Blue bars denote the mass concentrations of individual*
*elements/ionic species (left logarithmic axis, ng m⁻³) while the yellow points indicate the percent of species attributed to*
*the source (right axis).*

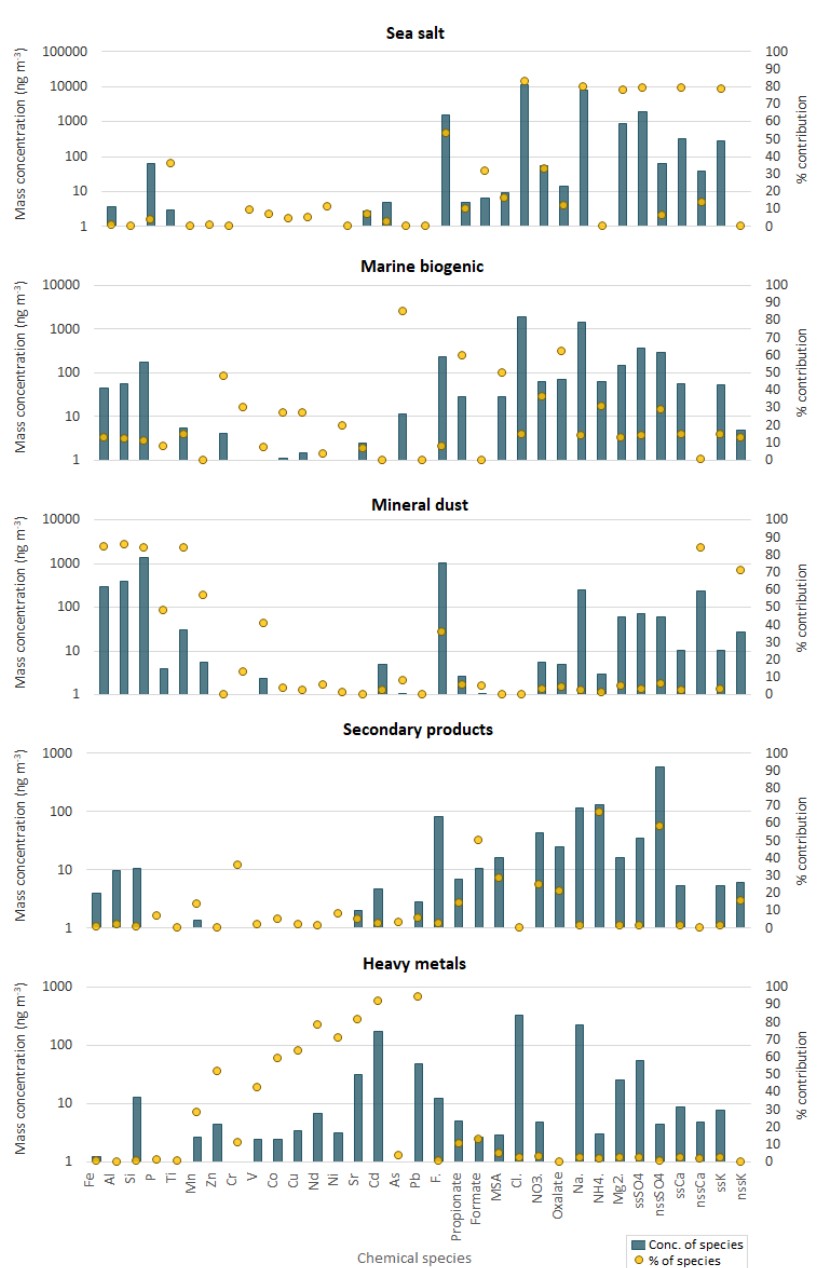
