# Peer review of "Chemical composition and source apportionment of atmospheric aerosols on the Namibian coast"

_Atmospheric Chemistry and Physics, 2020_

## Referee Comment (RC1) · Anonymous Referee #1 · 11 Jun 2020

The manuscript reports the first data on two years aerosol chemical composition at the Namibian site of Henties Bay Aerosol Observatory (HBAO). The site face the Atlantic Ocean, it present particular characteristic, a coastal desert which meteorology is affected by the upwelling of cold water of Benguela upwelling system and it deserve great interest. Besides, data on aerosol composition in this particular area are scarce, therefore the topic is relevant in the atmospheric scientific community and the paper deserve the publication on ACP. However, some implementation are necessary before the publication.

Specific comments

Line 148. The correlation SO4/S have to take into account the other species containing S (i.e. MSA) even if MSA presents lower concentration than SO4. It seems more correct report the correlation (MSA+SO4)/S, molar ratio will be more correct than weight ratio, even if MSA/S and SO4/S have only slightly different mass ratios.

Line 200. The Ca/Na w/w ratio in bulk seawater is 0.04 as correctly reported in table 2 and it is not 0.021.

Table 1. Looking at the values reported in this table I have few considerations:

There are surprising the very high maximum values for few species: Na, Cl and especially F. The STD DEV is not so high; therefore, these high values are occasional. Can be these anomalous high values due to contamination? Is it possible that something go wrong during the sampling? Only considering Na and Cl at their maximum, the sum is 129 ug/m3 that is really a huge PM10 mass value even for a marine windy environment.

It is still surprising 25 ug/m3 of fluoride, it is a huge values, it is difficult to have in open environment such as high concentration of F, this values is more similar to those found in the framework of health in dusty workplace. There are some other anomaly in the sample with the highest values of F, could be a problem of sampling or contamination? I do not believe these high values are reliable so please revise the sentence regarding these high values in the abstract and section 4.2.4.

In general, I suggest to e check the data for the sampling showing these se anomalous high values. Could these anomalous high values affect the PMF results?

There is another interesting feature of this data set: authors define this area desert-coastal, but looking at mean (but also maximum) values of crustal marker the concentration are quite low. For instance, Al present average value of only 478 ng/m3 and its maximum is only 4739 ng/m3. Therefore, the influence of sea spray aerosol is dominant respect to crustal aerosol, the author explain this by the wind intensity and

prevalent direction, that is correct, but this is surprising for this desert and arid region. I think this topic deserve a discussion.

Lines 366 and 370. Please check the year of these references.

Lines 452-458. About the seasonal cycle of MSA, figure 5 shows maxima in austral summer in 2017 (Oct-Nov-Dec) confirming the pattern already find in previous work, but in 2016 the maximum is in autumn (Mar-Apr), the author have an explanation for this maxima? Is this pattern anomalous or it is common to have a late phytoplanktonic bloom? In any case, they have to discuss the pattern that seems anomalous respect to previous results.

Equation 1b. As the authors have all the crustal element concentration, they can calculate the crustal content by the sum of the contributions of all the main crustal element oxides ($SiO_2$, $Al_2O_3$, $Fe_2O_3$, CaO, MgO, $K_2O$, $TiO_2$), following the approach reported in the literature by several authors (e.g. Marcazzan et al., 2001; Nava et al., 2012; Marconi et al., 2014) replacing CaO whit $CaCO_3$ basing on the Ca mineral content in the area. This approach can be more reliable than the use of the only Al and the averaged content of Al in the upper continental crust.

PMF

Here my main criticism to the paper. The PMF as it is do not add any new finding respect to the simply aerosol component analysis. Besides PMF is not able to distinguish the source of each metals, that are gathered in one factor called heavy metal. In my opinion, the PMF analysis is useless at this level. In the PMF, the author use the following not-independent variable: ss components (K, Ca and $SO_4$) and PM10 mass the latter obtained by the sum of the other aerosol component.

For this reason the ratio between sea spray components and the percentage of sea spray aerosol to the total mass are the same with or without PMF analysis. The sentence at lines 505-511 are obvious. In this context PMF do not add nothing new.

[Figure]

Line 526. Regarding the presence of As in this factor it has to be noticed that As concentration are really high to arise from marine biogenic activity, besides this factor is characterized by secondary species and As, because of it is a metal it is not secondary. Could be As is emitted by smelting activity and transported together with biogenic compounds? May air mass backward trajectory analysis for days with high As concentration clear this process?

Secondary product

This factor can be called NH4 neutralization, as it contain acidic species not necessary arising by the same source, but they are neutralized by ammonia (the latter find in aerosol phase as NH4+ counterbalancing HSO4-, MS-, Formate, oxalate etc.).

Heavy metals

Unfortunately, metals are gathered in only one factor, preventing the individuation of their source. There are several mining activity in Namibia (as reported in figure 1), therefore the analysis of backward trajectory for days with high concentration of each metals (or particular ratios between them) could be more useful than PMF in constrain metal sources.

Lines 627-631. This sentence is too general, I think in this upwelling area, nutrient in the ocean arise from sea bed by upwelling of water masses more than deposited from the atmosphere, I can be wrong but these sentence has to be better supported by literature.

References

Marcazzan, G. M., Vaccaro, S., Valli, G., and Vecchi, R.: Characterisation of PM10 and PM2.5 particulate matter in the ambient air of Milan (Italy), Atmos. Environ., 35, 4639–4650, 2001. Nava, S., Becagli, S., Calzolai, G., Chiari, M., Lucarelli, F., Prati, P., Traversi, R., Udisti, R., Valli G., and Vecchi R.: Saharan dust impact in central Italy: An overview on three years elemental data Records, Atmos. Environ., 60, 444–452,

2012. Marconi M., D. M. Sferlazzo, S. Becagli, C. Bommarito, G. Calzolai, M. Chiari, A. di Sarra, C. Ghedini, J. L. Gómez-Amo, F. Lucarelli, D. Meloni, F. Monteleone, S. Nava, G. Pace, S. Piacentino, F. Rugi, M. Severi, R. Traversi,and R. Udisti. Saharan dust aerosol over the central Mediterranean Sea: PM10 chemical composition and concentration versus optical columnar measurements. Atmos. Chem. Phys., 14, 2039–2054, www.atmos-chem-phys.net/14/2039/2014/ doi:10.5194/acp-14-2039-2014. 2014.
* * *

---

## Referee Comment (RC2) · Anonymous Referee #2 · 22 Jul 2020

[referee-annotated manuscript omitted]

---

## Author Response (AR1)

Atmos. Chem. Phys. Discuss.,
https://doi.org/10.5194/acp-2020-388-RC1

**Response to comments on "Chemical composition and source apportionment of atmospheric aerosols on the Namibian coast" by Danitza Klopper et al.**

The authors would like to thank the two anonymous referees for providing a thoughtful and constructive review on the manuscript. We considered their comments with the highest possible attention.

This concerned in particular the remarks on the PMF analysis. Both Referees #1 and #2 agreed on the fact that the PMF was not bringing sufficiently highlights and additional information with respect to the discussion from the chemical apportionment, and because of that, did not deserve a dedication section of the manuscript. While agreeing on that, we took the advantage of the referee's suggestions to

- Revise our uncertainty analysis on the elemental concentrations by XRF (section 2.1, comment of Referee#2), and therefore revise the input error matrix in PMF (section 3.2);

- Perform sensitivity studies of the PMF including or excluding peak concentrations of As, F-, Na and Cl- (comment of Referee#1);

- Discarding the separation of the sea salt and non-sea salt components of $SO_4^{2-}$, $Ca^{2+}$ and $K^+$ (comment of Referee#1);

Those analyses resulted in a new and more robust PMF solution, which provides additional information with respect to the chemical apportionment analysis. In particular, the obtained new PMF solution provides with the source mass apportionment, distinguishing natural and anthropogenic sources, and furthermore apportions the heavy metals into distinguished sources related to mining and smelting/ship traffic activities. These new results are presented in the abstract and the conclusions of the manuscript (section 5), as well as in section 4.2, which is now called "Variability and apportionment of measured concentrations". Section 4.3 ("Source apportionment") of the original manuscript is now suppressed.

Additional major modifications are

- Elemental/ionic ratios for the *sea salt* and the *mineral dust* PMF components are now shown in Tables 2 and 3
- To improve clarity, a number of tables and figures were added to the supplementary material:
  - Table S1. Correction factor used to scale up the elemental concentrations measured by XRF to account for the X-ray self-attenuation effects in the individual particle grains. A mean diameter of 4.5 μm is chosen to represent the average coarse particle size.
  - Figure S3. PMF mass apportionment for the five component solution.
  - Figure S4. Bivariate polar plots for (a) vanadium and (b) nickel, showing the variability in mean concentrations with changes in wind speed and direction. Wind direction is indicated by the cardinal point in the four quadrants, mean wind speed (m s-1) is indicated by the concentric circles from the centre of the plot and the mean concentrations are measured in ng m-3, and given by the gradient colour scale.
  - Figure S5. Pie charts of the PMF mass apportionment of V and Ni measured at HBAO. Legends provide with the name of the source component and the fraction of the contributed mass elemental concentration.
  - Figure S6. Scatterplot of F- with respect to nssCa2+ for 2016 (blue) and 2017 (orange). Concentrations are expressed in μg m-3. The slope and the Pearson correlation coefficient (R2) are indicated.
  - Figure S7. Same as Figure S5 for F- concentrations.

Additional comments from Referees #1 and 2 are addressed here below (comments by referees in bold and our accompanying responses in plain text), as well as the text revision (in blue) are represented hereafter.

Incorporating the suggestions from both referees has certainly improved the manuscript which we hope is now acceptable for publication in ACP.

**Referee #1: specific comments**

**Line 148. The correlation SO4/S have to take into account the other species containing S (i.e. MSA) even if MSA presents lower concentration than SO4. It seems more correct report the correlation (MSA+SO4)/S, molar ratio will be more correct than weight ratio, even if MSA/S and SO4/S have only slightly different mass ratios.**

The aim here was to check the correlation of measurements from ion-chromatography and XRF. We agree with the referee that it would also be correct to report the molar ratio, specifically with the sulphur containing MSA and sulphate species. Subsequently, the molar ratios for $MSA+SO_4^{2-}/S$ were added to the discussion for a more complete representation and the text was updated in lines 154 - 158 as follows, "Mass ratios were 1.3 ± 0.1 (2016) and 1.0 ± 0.1 (2017), 1.3 ± 0.1 (2016) and 0.9 ± 0.1 (2017), and 2.0 ± 0.1 (2016) and 1.7 ± 0.2 (2017) for $Cl^-/Cl$, $Na^+/Na$, and $Mg^{2+}/Mg$, respectively. Conversely, no annual dependence was observed in the slopes of the linear correlations for $Ca^{2+}/Ca$ ($\sim$ 0.8 ± 0.1), $K^+/K$ ($\sim$ 0.6 ± 0.1) and $MSA+SO_4^{2-}/S$ ($\sim$ 2.7 ± 0.4). The molar ratio of $(MSA+SO_4^{2-})/S$ was 8.0 ± 1.2 for 2016 and 7.8 ± 0.9 for 2017". The scatterplot of $SO_4^{2-}/S$ in Figure S1 of the supplementary material was updated to include $MSA+SO_4^{2-}/S$.

**Line 200. The Ca/Na w/w ratio in bulk seawater is 0.04 as correctly reported in table 2 and it is not 0.021.**

We thank the referee for noticing this typographic mistake. It has been corrected and all other reported values were double-checked for such errors. No other issues were found.

**Table 1. Looking at the values reported in this table I have few considerations:**

**There are surprising the very high maximum values for few species: Na, Cl and especially F. The STD DEV is not so high; therefore, these high values are occasional. Can be these anomalous high values due to contamination? Is it possible that something go wrong during the sampling? Only considering Na and Cl at their maximum, the sum is 129 ug/m3 that is really a huge PM10 mass value even for a marine windy environment. It is still surprising 25 ug/m³ of fluoride, it is a huge values, it is difficult to have in open environment such as high concentration of F, this values is more similar to those found in the framework of health in dusty workplace. There are some other anomaly in the sample with the highest values of F, could be a problem of sampling or contamination? I do not believe these high values are reliable so please revise the sentence regarding these high values in the abstract and section 4.2.4. In general, I suggest to e check the data for the sampling showing these se anomalous high values. Could these anomalous high values affect the PMF results?**

We have considered in great detail the reliability and uncertainties of the values of highest concentrations of Na, Cl and F which occurred during the sampling period. As the referee does not specify his/her criteria concerning which values should be considered as "very high maximum values ", we explored this issue by qualifying these values as those concentrations exceeding the arithmetic mean plus twice the standard deviation of the population (mean + 2 x STD DEV). The resulting "very high maximum values" are 23.9, 33.7 and 9.8 µg m$^{-3}$ for $Na^+$, $Cl^-$ and $F^-$ respectively.

Concerning Cl⁻ and Na⁺, 11 samples exceeded this lower limit out of the total 385 values available (Cl⁻ and Na⁺ mass concentrations were above the MQL on all the sampled filters) Dates of those 11 occurrences are provided in Table A1.

***Table A1.*** Start date and time for sampling corresponding to values exceeded the "very high maximum values" of Na⁺ and Cl⁻, and F⁻.

| Start date and time (UTC) | | |
|---|---|---|
| **Na⁺ and Cl⁻** | **F⁻** | |
| 28-02-2016 7:00 | 27-02-2016 19:00 | 29-08-2016 20:00 |
| 01-04-2016 19:00 | 28-02-2016 7:00 | 25-03-2017 19:00 |
| 03-04-2016 8:00 | 21-03-2016 7:00 | 06-10-2017 21:00 |
| 04-04-2016 8:00 | 06-04-2016 8:00 | 15-11-2017 21:00 |
| 05-04-2016 8:00 | 06-04-2016 20:00 | 16-11-2017 9:00 |
| 06-04-2016 8:00 | 08-04-2016 8:00 | 16-11-2017 21:00 |
| 28-08-2016 8:00 | 28-08-2016 8:00 | 17-11-2017 9:00 |
| 28-08-2016 20:00 | 28-08-2016 20:00 | 17-11-2017 21:00 |
| 29-08-2016 8:00 | 29-08-2016 8:00 | |
| 12-10-2016 19:00 | | |
| 01-11-2017 9:00 | | |

Besides three occasional occurrences, values of Cl⁻ and Na⁺ above the set threshold occurred during two specific periods, from 1 to 6 April 2016 and from 28 to 29 August 2016. The corresponding wind speed for these 11 samples was moderate (arithmetic mean 3.2 ± 0.9 m s⁻¹) and comparable to the average values during the sampling campaign with these 11 values excluded (3.8 ± 1.7 m s⁻¹). Polar plots for Na⁺ and Cl⁻, shown in Figure A1, indicate that the highest mean concentrations were associated to sea breezes from the SSE and WNW, also characteristic of the year-round mean wind direction. The mean mass ratio of Cl⁻/Na⁺ for the 11 samples (1.4 ± 0.1) is consistent with the mean mass ratio for the data with these values excluded (1.3 ± 0.1), suggesting no specific analytical issue. For these reasons, we suggest that the 11 samples of Cl⁻ and Na⁺ do not present anomalously high concentrations and can be retained in further analyses.

Regarding F-, and as explicitly stated in the manuscript (section 4.2.4), we agree with Referee#1 that the mass concentration values measured at the site are unexpectedly high and more typical of a polluted work place more than a semi-remote environment. The mean mass ratios of F⁻/Na⁺ for 2016 and 2017 (0.38 ± 0.24 and 0.32 ± 0.35, respectively) are orders of magnitude higher than expected for marine sea salt. The enrichments were attributed to mining and road construction activities as a consequence of the enrichment of fluoride in the Namibian soils, which is also suggested by the excellent correlation between F⁻ and nss-Ca²⁺, which is now shown in the supplementary material (Figure S6). The F6 source apportionment is shown in Figure S7. Section 4.2.4 was partially rewritten as (lines 486-501) "The very good correlation of F- with nss-Ca2+, shown in Figure S6 (R2 equal to 0.76 in 2016 and to 0.84 in 2017), yielded a mean mass ratio of 6.4 and 5.8, respectively, much higher than reported in groundwater, aerosols or precipitation in polluted environments (Feng et al., 2003; Prodi et al., 2009).

The strong relationship to nss-Ca2+ (and a posteriori to Ca2+) drove the PMF apportionment (Figure S7), which attributed approximately the 94% of the F- mass concentrations to the sea salt and mineral dust components (55.1 ± 1.9% and 38.8 ± 1.1%, respectively), and the remaining 6% to fugitive dust (2.3 ± 0.5%) and industry (3.8 ± 1.0%). Possible sources are the emission of fugitive dust during fluorspar mining of carbonatite related fluorospar deposits at the Okorusu Mine (20°3'S, 16°44'E), but very likely also from the periodic surface mining occurring approximately 20 km south of HBAO to provide gravel for the construction of a major road between Swakopmund and Henties Bay which started late in 2015 (A. Namwoonde, pers. corr.). The evaporation of fluoride rich water, leached into groundwater (Wanke et al., 2015, 2017) from fluoride-rich mineral deposits and soils, throughout the region and in the coastal waters (Compton and Bergh, 2016; Mänd et al., 2018), would also increase atmospheric F- concentrations. In an analysis of borehole water in Namibia, roughly 80% of those sites surveyed were deemed unsafe to drink as a direct result of high fluoride concentrations (Christelis and Struckmeier, 2011). "

Furthermore, we have given additional attention to the "very high maximum values". For $F^-$, this condition was met by 17 samples (out of 385, Table A1). As for $Cl^-$ and $Na^+$, the occurrences of extreme values of $F^-$ are not isolated but corresponded to a few-day long periods. The polar plot of $F^-$ mass concentrations with respect to wind direction (Figure A1c) shows that these values corresponded to south-easterly winds exceeding 8 m s$^{-1}$, which could favour dust resuspension.

(a)             (b)             (c)

[Figure]

**Figure A1.** Bivariate polar plots for (a) Na+, (b) Cl- and (c) F- showing the variability in mean concentrations with changes in wind speed and direction. Wind direction is indicated by the cardinal point in the four quadrants, mean wind speed (m s$^{-1}$) is indicated by the concentric circles from the centre of the plot and the mean concentrations are measured in ng m$^{-3}$, and given by the gradient scale on the right of plot.

**1.**    **The mean F-/nssCa$^{2+}$ mass ratio for those occurrences is 6.8± 2.5, close to the annual means, suggesting the validity of these values for further analysis.**

Nevertheless, we agree with Referee#1 that in principle the PMF solution could be biased by those very high concentrations for F-, Cl- and Na+. To explore this possibility, an additional PMF simulation was performed after removing the peak F- values, and then another after removing the peak Cl- and Na+ values. The results of the original PMF with these values included and with them excluded are given in Table A2 below.

The comparison of the two simulations indicates that the inclusion of the F- "very high maximum values" only marginally modify the results of the PMF simulations but not the identification of the factors which is maintained and prove the robustness of the solutions. The same is true for the PMF solution with the highest Na+ and Cl- values removed. The largest change (but marginal) when discarding the highest F- values is the lowered F- to the *mineral dust* component and the increased contribution to the *sea salt* and *marine biogenic* components, which entrains a lowered contribution of the *mineral dust* component to the total mass concentration. In the case of Na+ and Cl-, changes in the contributions of the different components to these species were smaller than 2% with the biggest differences in *sea salt* and *marine biogenic* components. These differences were not statistically significant and the "very high maximum values" were retained in the dataset.

**Table A2.** PMF solutions for "very high maximum values" of F⁻, Cl⁻ and Na⁺, retained in the dataset, with the original uncertainty file, compared with the solution where "very high maximum values" of F⁻ were removed from the dataset.

| With peak values left in the dataset | With peak F⁻ values removed from the dataset | With peak Cl⁻ and Na⁺ values removed from the dataset |
|---|---|---|

[Figure]

| Contribution of the components to the F⁻ mass concentration | | Contribution of the components to the F⁻ mass concentration | | |
|---|---|---|---|---|
| mineral dust | 34.2 ± 0.1% | mineral dust | 28.1 ± 0.1% | |
| sea salt | 56.2 ± 0.1% | sea salt | 58.8 ± 0.1% | |
| marine biogenic | 4.7 ± 0.4% | marine biogenic | 7.2 ± 0.4% | |
| heavy metals | 1.3 ± 5.3% | heavy metals | 1.3 ± 2.8% | |
| ammonium neutralised | 3.6 ± 1.4% | ammonium neutralised | 3.3 ± 0.4% | |

| Contribution of the components to the mass concentration of $Na^+$ and $Cl^-$, respectively | | | | Contribution of the components to the mass concentration of $Na^+$ and $Cl^-$, respectively | | |
|---|---|---|---|---|---|---|
| mineral dust | 2.9 ± 0.4% | 0 ± 0.3% | | mineral dust | 2.8 ± 0.5% | 0 ± 0.4% |
| sea salt | 81.6 ± 2.5% | 85.3 ± 2.3% | | sea salt | 80.0 ± 2.5% | 83.6 ± 2.2% |
| marine biogenic | 13.7 ± 0.2% | 14.1 ± 0.1% | | marine biogenic | 15.3 ± 0.7% | 15.6 ± 0.7% |
| heavy metals | 0.5 ± 0.3% | 0.6 ± 0.2% | | heavy metals | 0.7 ± 0.2% | 0.8 ± 0.3% |
| ammonium neutralised | 1.2 ± 0.8% | 0 ± 0.9% | | ammonium neutralised | 1.2 ± 0.3% | 0 ± 0.3% |

| Contribution of the components to the total mass concentration | | | | | |
|---|---|---|---|---|---|
| mineral dust | 10.6 ± 0.1% | mineral dust | 8.3 ± 0.1% | mineral dust | 10.4 ± 0.2% |
| sea salt | 71.7 ± 0.1% | sea salt | 70.6 ± 0.1% | sea salt | 69.5 ± 2.0% |
| marine biogenic | 12.8 ± 0.2% | marine biogenic | 15.5 ± 0.2% | marine biogenic | 15.4 ± 0.6% |
| heavy metals | 1.5 ± 1.0% | heavy metals | 2.4 ± 0.8% | heavy metals | 1.3 ± 0.3% |
| ammonium neutralised | 3.5 ± 0.4% | ammonium neutralised | 3.7 ± 0.2% | ammonium neutralised | 3.5 ± 0.3% |

**There is another interesting feature of this data set: authors define this area desert coastal, but looking at mean (but also maximum) values of crustal marker the concentration are quite low. For instance, Al present average value of only 478 ng/m3 and its maximum is only 4739 ng/m³. Therefore, the influence of sea spray aerosol is dominant respect to crustal aerosol, the author explain this by the wind intensity and prevalent direction, that is correct, but this is surprising for this desert and arid region. I think this topic deserve a discussion.**

We can see how the term "coastal desert" can be confusing considering the high variability of surface cover in Namibia. The landscape is primarily covered by loose sand in dune fields, but also gravel plains and few clay sources, where aluminosilicates would be sourced from. To clarify, the text (line 357 - 362) was updated as follows, "Our arid sampling site is surrounded by loose sand, gravel plains (Matengu et al., 2019) and the deep Omaruru river valley directly north of the sampling site which is also a recognised source of mineral dust to the offshore waters (Tlhalerwa et al. 2012). While mostly characterised by gravels, some clay-rich deposits are found around the river valley approximately 17 km northeast of HBAO (Matengu et al., 2019). The relatively low aluminium concentrations measured at HBAO suggest that these are not a major local source for the site."

**Lines 366 and 370. Please check the year of these references.**

The dates for these references were checked and corrected, thank you. All other references were also checked.

**Lines 452-458. About the seasonal cycle of MSA, figure 5 shows maxima in austral summer in 2017 (Oct-Nov-Dec) confirming the pattern already find in previous work, but in 2016 the maximum is in autumn (Mar-Apr), the author have an explanation for this maxima? Is this pattern anomalous or it is common to have a late phytoplanktonic bloom? In any case, they have to discuss the pattern that seems anomalous respect to previous results.**

The MSA variability and its links to the phytoplankton blooms is certainly an issue that cannot be solved today, because of the lack of direct measurements of the bloom spatial distribution. Unfortunately, satellite images are rare due to the persistent cloud cover. The MSA seasonal cycle was already discussed in Formenti et al. (2019). The text in the manuscript was updated as follows (lines 535-546) "The MSA concentrations measured at the site ranged between 10 and 230 ng m-3 (Table 1). The mean annual concentration was 63 ± 39 ng m-3, three times higher than the mean value of 20 ± 20 ng m-3 (6.2 ± 4.2 ppt) reported by Andreae et al. (1995) over the open ocean along 19°S, and lower than in the southeast Atlantic Ocean (Zhang et al., 2010; Table 4). As already described in Formenti et al. (2019), the MSA concentrations were higher in the austral summer and spring and lower in the austral winter. DMS is more efficiently oxidised in warmer conditions (Ayers et al., 1997; Huang et al., 2017) which explains the the higher daytime mean concentrations of marine biogenic products (MSA and nss-SO42-) and lower means at night and in the winter. Springtime averages for MSA were in the range of that measured by Huang et al. (2018) during a springtime cruise over the South Atlantic and by Prodi et al. (2009) in the Venice Lagoon (Table 4). The mismatch of seasonality with respect to that of the phytoplankton blooms (Louw et al., 2016) is already discussed by Formenti et al. (2019) and attributed to the spread of blooms in the BUS region depending on local conditions."

**Equation 1b. As the authors have all the crustal element concentration, they can calculate the crustal content by the sum of the contributions of all the main crustal element oxides (SiO2, Al2O3, Fe2O3, CaO, MgO, K2O, TiO2), following the approach reported in the literature by several authors (e.g. Marcazzan et al., 2001; Nava et al., 2012; Marconi et al., 2014) replacing CaO whit CaCO3 basing on the Ca mineral content in the area. This approach can be more reliable than the use of the only Al and the averaged content of Al in the upper continental crust.**

Our choice of the simplified equation with respect to the more complete one (sum of main crustal element oxides as SiO2, Al2O3, Fe2O3, CaCO3, MgO, K2O, TiO2) suggested by Referee#1 is motivated by the fact that the Si/Al ratios measured at HBAO are ~3.6 for both years (see Table 3 in the paper) and close to those expected for the upper continental crust (see Rudnick, R.L. and S. Gao (2003). 3.01 - Composition of the Continental Crust. Treatise on Geochemistry. H. D. Holland and K. K. Turekian. Oxford, Pergamon: 1-64.) than for airborne mineral dust (Table 3 and references therein). This is consistent with the fact that the Namibian soils are characterised by granite and sand banks that dominate over clay deposits (Matengu et al., 2019).

To illustrate the impact of our choice on the evaluation of the estimated dust mass (EDM), Figure A2 illustrates the comparison between the EDM by two approaches, the Referee#1 equation on the y-axis and Equation 1.b in the manuscript on the x-axis.

[Figure]

*Figure A2.* Scatterplot for the estimated dust mass using the Referee#1 equation to that estimated by Equation 1.b in the paper.

The comparison shows that indeed, on average, our simplified equation overestimates the EDM calculated with the Referee#1 equation by 18% in 2016 and 7% in 2017. The underestimation is understandable when comparing the terms of the largest difference which the contribution of aluminosilicate and quartz in the upper continental crust (and accounted for as Al/0.0813) with respect to that obtained through the Referee#1 equation (1.89*Al+2.14*Si), which results in a Si/Al ratio of 1.13. On the other hand, the use of either equation does not induce any temporal bias to the analysis as the linear correlation of the estimated dust masses by the two approaches is excellent ($R^2$= 0.98). In conclusion, we consider that the estimated dust mass in our approach (Equation 1b) is more adapted as a term of comparison for the PMF analysis applied to the Namibian aerosols.

However, for sake of clarity, in re-organising the manuscript, we have now suppressed those comparisons from the main text.

**PMF**

**Here my main criticism to the paper. The PMF as it is do not add any new finding respect to the simply aerosol component analysis. Besides PMF is not able to distinguish the source of each metals, that are gathered in one factor called heavy metal. In my opinion, the PMF analysis is useless at this level. In the PMF, the author use the following not-independent variable: ss components (K, Ca and SO4) and PM10 mass the latter obtained by the sum of the other aerosol component. For this reason, the ratio between sea spray components and the percentage of sea spray aerosol to the total mass are the same with or without PMF analysis. The sentence at lines 505-511 are obvious. In this context PMF do not add nothing new.**

Please see general comments regarding the significance of the PMF results.

The PMF solution with sea salt and non-sea salt fractions of $SO_4^{2-}$, $Ca^{2+}$ and $K^+$ already separated (as presented in the paper) was compared with the solution where these values were replaced by independent variables $SO_4^{2-}$, $Ca^{2+}$ and $K^+$. Table A3 shows the comparison of the sea salt fractions as evaluated in the two approaches (note that the two simulations yielded a 5 components as the best solution).

*Table A*Erreur ! Il n'y a pas de texte répondant à ce style dans ce document.*2.* The sea salt fractions of $SO_4^{2-}$, $Ca^{2+}$ and $K^+$ when estimated by the (i) PMF where the ss and nss components were already separated in the input dataset, (ii) PMF where these components were not separated (i.e. independent values only), and (iii) ratios to unique tracers.

|  | Sea salt component (%) | | |
|  | PMF with ss and nss components separated | PMF with independent values only | Ratios to unique tracers |
| --- | --- | --- | --- |
| $SO_4^{2-}$ | 61.9 | 66.6 | 57.0 |
| $Ca^{2+}$ | 49.0 | 53.0 | 44.5 |
| $K^+$ | 74.6 | 75.1 | 89.0 |

The results for the two PMF solutions are in good agreement (within 5%), and in agreement (within 10%) with the proportions estimated by ratios to unique tracers. A section was added to the text in lines 339 - 340 as follows, "The PMF estimated that sea salt contributed to 53.0 ± 1.6% of the calcium and 75.1 ± 2.4% of the $K^+$."

We also mentioned the PMF separation of sulphates in line 527, as follows, "The PMF estimated that the *sea salt* component contributed to 66.6 ± 0.4% of the sulphate mass."

The old PMF solution (ss and nss components separated out) attributed some of the nss portions of the $SO_4^{2-}$ and $Ca^{2+}$ to the *sea salt* component and large fractions of the ss portions to components other than the *sea salt* source. This would change the contribution to the total PM10 mass, as noted by the referee, and the separation of the species to the different PMF components as compared to when we use only the independent values. In the interests of keeping the results of the ratios to unique tracers and PMF separate, the referee's suggestion to present the PMF with independent variables was accepted.

The uncertainty file used in the PMF was also updated after consideration of the comments from Referee #2. The text related to the PMF results was updated accordingly to reflect the calculation of the uncertainty file (line 235 - 238) as follows, "In order to weight the concentrations according to their amount, a relative uncertainty of 10%, 20% and 60% was attributed to each value of concentration in the input matrix based on their ratio to their respective MQL (larger than 3.3, comprised between 1.25 and 3.3, and comprised between 1 and 1.25, respectively).." The results were updated throughout the document along with Figure 6 in the paper.

The five component solution obtained by PMF with this new input concentration file (with independent values for $SO_4^{2-}$, $Ca^{2+}$ and $K^+$ only) and uncertainty file (with a higher relative uncertainty) was different from that obtained originally. In fact, this new solution addresses the issue as raised by the referee, by separating the heavy metals into two components.

**Line 526. Regarding the presence of As in this factor it has to be noticed that As concentration are really high to arise from marine biogenic activity, besides this factor is characterized by secondary species and As, because of it is a metal it is not secondary. Could be As is emitted by smelting activity and transported together with biogenic compounds? May air mass backward trajectory analysis for days with high As concentration clear this process?**

The referee is correct in saying that these values for As are high, and that there is no discernible correlation between As and MSA, as discussed in the text. After the inclusion of the new uncertainty file and the removal of the already separated ss and nss components, the PMF solution now associates the As to other heavy metals (Zn, Cu, Ni and Sr) and not to the MSA and other secondary products as in the original PMF solution as reported in the paper. This PMF component is characteristic of smelting operations and oil combustion and was therefore labelled *industry.* The discussion regarding this component can be found in section 4.2.5.

To evaluate the very high As concentrations, the threshold limit was calculated in the same way as for $Cl^-$, $Na^+$ and $F^-$, as the mean + 2 x STD DEV (825 ng m$^{-3}$).

Measurements made during two sampling weeks (24-31 January and 8-15 February 2017) were classified by these very high As concentrations. In order to better differentiate between the sources of air masses arriving during these two weeks, trajectories and local winds were analysed and are given in Figures A3 and A4. These results suggest primarily marine regions as the source of these air masses, although resolutions are very coarse.

[Figure]

**Figure A3.**   Back-trajectories run for 72 hours (each point representing 6 hours), starting for each sampling period when arsenic concentrations were greater than 2 standard deviations above the mean (825 ng m$^{-3}$). Each 72-hour back-trajectory is presented by a different colour.

[Figure]

**Figure A4.** Wind roses of local winds measured at HBAO during sampling where arsenic concentrations were > 825 ng m$^{-3}$.

**Secondary product**

**This factor can be called NH4 neutralization, as it contain acidic species not necessary arising by the same source, but they are neutralized by ammonia (the latter find in aerosol phase as NH4+ counterbalancing HSO4-, MS-, Formate, oxalate etc.).**

This is a great suggestion, thank you. The name was changed to *ammonium neutralised*.

**Heavy metals**

**Unfortunately, metals are gathered in only one factor, preventing the individuation of their source. There are several mining activity in Namibia (as reported in figure 1), therefore the analysis of backward trajectory for days with high concentration of each metals (or particular ratios between them) could be more useful than PMF in constrain metal sources.**

Yes, you are correct and the new PMF solution (after consideration of the "main criticism to the paper") now separates the heavy metals into two components, namely *industry* and *fugitive dust*. These results were now incorporated into sections 4.2.,4.2.3. and 4.2.5.

For completeness we also ran back-trajectories corresponding to the sampling times during the week of 6 – 13 October 2016, when the overall highest concentrations of all heavy metals were measured (except for As). Figure A5 shows that primarily marine air masses that were transported southerly within the coastal margin and over, eg. the heavy-oil fuelled power plants or industry, and over a preferential commercial shipping transport pathway (Cape of Good Hope sea route), as mentioned in the paper (lines 106).

[Figure]

*Figure A5.* Back-trajectories run for 72 hours (each point representing 6 hours), starting at the same time as the end of the filter sampling during the week of 6 to 13 October 2016. Each 72-hour back-trajectory is presented by a different colour.

Back-trajectories were also run for additional episodes when high V concentrations were measured. These high V concentrations were classified in the same way as for As and F-. These showed the arrival of primarily continental air masses. Due to the well-mixed atmospheric composition and the coarse resolution of trajectory input data, specific heavy metal contributions from sources, such as the multitude of mines to the northeast of HBAO, could not be distinguished from one another on this basis. This is discussed in section 5, lines 615-618, "While the coarse resolution of air mass backtrajectories and the dominance of marine air masses does not allow to distinguish sources at the country scale, the PMF

analysis performed in this paper was able to identify the specific and distinct contribution of mining activities, including for road construction for the majority of the heavy metals (ex., V).”

**Lines 627-631. This sentence is too general, I think in this upwelling area, nutrient in the ocean arise from sea bed by upwelling of water masses more than deposited from the atmosphere, I can be wrong but these sentences have to be better supported by literature.**

Referee#1 is right, the Benguela upwelling region is likely not very sensitive to atmospheric input of macronutrients, which, however, could be important for the productivity of the near-coast waters or, conversely, farther downwind towards the Southern Atlantic where the oceanic upwelling is not active.

To improve its clarity, the sentence was rewritten and updated in line 622 - 630 as follows “The deposition of macronutrients (P, Fe..) from the outflow of mineral dust is not expected to be relevant for the BUS region, one of the most productive marine environments in the world, while it could be important in fertilising waters near the coast (Dansie et al., 2017) and in the Southern Ocean (Okin et al., 2011). On the other hand, the atmospheric deposition of trace metals (Cr, Cu, Ni, Mn, or Zn) in the aerosols, which play a biological role in enzymes and as structural elements in proteins (Morel and Price, 2003), could affect the marine productivity of the BUS and should be explored in future work. The complexity and diversity of sources that might contribute to the mineral dust load at HBAO over a year, as well as the detailed chemical composition including trace metal contamination, deserve certainly further dedicated investigation”.

**Referee #2: specific comments**

**Line 120: This is likely true for elements with high concentration. I wonder that trace-elements, often just above the quantification limit, could be detected with negligible statistical uncertainty.**

Referee#2 is correct in indicating that, in spite of its importance, the systematic error due to the calibration of the XRF machine does not represent the full analytical uncertainties on the final elemental concentrations. Additional terms contributing to it are

- The uncertainty related to the uniformity of the aerosol deposit on the filters, and the scaling error that can occur due to the fact that the area of the deposit which is analysed is smaller than the area of the aerosol deposit;

- The statistical error on the photon counts, in particular for trace elements whose concentrations are close to their detection limits.

- For the lightest elements (Z < 20), the choice of the correction factor to account for the self-attenuation of the X-ray signal, in particular for particles larger than 1 µm in diameter (Formenti et al., 2010).

These sources of errors have been carefully investigated through the years, and many precautions have been taken at LISA in the construction of a decadal experience in XRF analysis. Former analyses, often unpublished, have supported this experience, and provided with mitigation strategies, while not always resulted into their exact quantification. For example, in order to improve statistics and reduce the statistical error on the photon counts for trace elements, we systematically repeat each analysis three times.

To take all these considerations into account, we have therefore revised our error budget and attributed a 10% uncertainty to all elemental concentrations presented in the paper. The text (lines 121 – 132) was updated as follows

"Elemental concentrations of 24 elements (Na, Mg, Al, Si, P, S, Cl, K, Ca, Ti, V, Cr, Mn, Fe, Co, Ni, Cu, Zn, As, Sr, Pb, Nd, Cd, Ba) were obtained at LISA by wavelength-dispersive X-ray fluorescence (WD-XRF) using a PW-2404 spectrometer (Panalytical, Almelo, Netherlands), according to the protocol previously described by Denjean et al. (2016). The relative analytical uncertainty on the measured atmospheric concentrations (expressed in ng m$^{-3}$) is evaluated as 10%. This represents the upper limit uncertainty, taking into account:

- The uncertainty related to the uniformity of the aerosol deposit on the filters, and the scaling error that can occur due to the fact that the area of the deposit which is analysed is smaller than the area of the aerosol deposit;

- The statistical error on the photon counts, in particular for trace elements whose concentrations are close to their detection limits;

- The percent error on the certified mono- and bi-elemental standard concentrations (Micromatter Inc., Surrey, Canada) used for calibration of the XRF apparatus;

- For the lightest elements (Z < 20, Na to Ca), the choice of the correction factor to account for the self-attenuation of the X-ray signal, in particular for particles larger than 1 µm in diameter (Formenti et al., 2010). Constant correction factors (Table S1) were estimated through the sampling period assuming a mean diameter of 4.5 µm to represent the average coarse particle size."

**Line 122 – 123: Such average correction introduces a further term in the uncertainty budget. A quantification should be included in the text.**

As shown in Formenti et al. (2010), the self-attenuation correction depends on the individual particle size as well as on its composition. The information on the particle size at HBAO was only partially available during the period of sampling. Particle size was measured by Aerodynamic Particle Sizer (APS) which experienced very high losses for particles larger than 2.5 µm in diameter, therefore could not be used to inform on the full extent of the size distribution of the marine aerosols. The particle size was measured by a GRIMM optical counter during the shorter period of the AEROCLO-sA field campaign (Formenti et al., 2019). Figure A6 illustrates snapshots of these measurements (3-minute averages at different wind speed conditions) and the importance, as expected, of the coarse mode.

[Figure]

***Figure A6.*** Aerodynamic Particle Sizer measurements (3-minute averages at different wind speed conditions) during the AEROCLO-SA field campaign.

A mean diameter of 4.5 µm was chosen to represent the average coarse particle size to evaluate the self-attenuation correction. For the sake of completeness, Table S1, with the correction factors used to scale up the elemental concentrations measured by XRF, was added to the supplementary material.

**Line 146: I see a possible discrepancy for Na only: being this element the most sensitive to self-attenuation effects in the XRF-analysis, a reason for the difference could be related to the mean dimension of the sea-salt particles in the two period.**

In principle, Referee#2 is right. However, changes in the $Na^+$/Na slopes in 2016 occur per batch of analysed data, which points out to some differences in the analyses, and not at a higher frequency, as it would be expected as a result of changes in the particle size distribution during sea spray emission. This is also in accordance with the fact that the $Cl^-$/Cl slope varied in the same way with time (higher in 2016 and lower in 2017). As we said when replying to the previous comment, unfortunately we only have very partial information regarding the aerosol size distribution at HBAO from the measurements of the APS. From those measurements, the aerosol size distribution for particles smaller than 2.5 µm in diameter held rather constant with time.

**Line 154: again, this could be due to the self-attenuation effect and to the choice of the corrective factor**

Agreed. The sentence was modified at lines 161 - 165 to reflect this as follows "No specific sampling nor analytical problems were found. However, the further comparison of their proportions to those expected for seawater (Seinfeld and Pandis, 2006) as well as the possibility that the choice of a mean, time-independent self-attenuation correction factor, would be erroneous, at least for Na and Mg, suggested to discard the XRF results and only use the values obtained by IC for those three elements."

**Line 180: by receptor models (i.e. Positive Matrix Factorization, PMF)**

Corrected, thank you.

**Line 184: I'd not define sea-salt or resuspension as "emission " sources. I'd also write "source" between quotation marks.**

The referee is correct and subsequently we chose to define them as "source types" which offers a more inclusive description. Where applicable "source" or "factor" were replaced with PMF "components" for clarity. Remaining references to "sources" identified by the PMF were put in quotations.

**Line 209: Factorization**

For consistency with British English throughout the document, the format "factorisation" was kept.

**Line 299: actually 10 +/- 4, Line 304: 1.3 +/- 0.1, Line 552: (3.2 +/- 1.0) % - (all factor mass %)**

Significant digits were corrected, thank you.

**Line 366: actually, there are many other previous papers indicating the V:Ni ratio as tracer of heavy oil combustion**

The referee is correct and we have added three references to lines 431 - 434, thank you.

**Line 372: The value of the slope of such correlation curve should be given and discussed in comparison with the usual literature figure, i.e. V:Ni ca = 3.  This is also related to the discussion from line 380 on**

As V and Ni were only poorly correlated, the annual average V/Ni ratios for 2016 and 2017 were preferred to the slope of the correlation curve. These are already reported in the manuscript (lines 430 - 433) and the paragraph was updated to include the PMF results, stating, "The V/Ni ratio for 2016 is 1.7 ± 1.1 and 2017 is 1.3 ± 1.3, and 0.5 ± 0.1 for the PMF *industry* component, lower than reported by Lyyränen et al. (1999) and Corbin et al. (2018) for heavy fuel oil in diesel engines, and by Becagli et al. (2017) and Viana et al. (2009) in the Mediterranean basin ambient air (2.8–2.9 and 4–5, respectively)."

At lines 434 - 435 we included, "…moderate to good correlations of V and Ni with Zn (0.42 and 0.55, respectively), Cu (0.55 and 0.73) and Pb (0.56 and 0.69)…"

We also included "…poor correlation ($R^2$ around 0.3)." at line 426.

**Line 489: I suggest to write the name of the five sources in italic. Furthermore, I do not see appropriate the use of "source" to identify those which appear to be "components" of the PM, not always directly linked with a specific process (e.g. heavy metal, secondary products)**

Considering these suggestions and comments from Referee #1, all references to "PMF sources" were changed to "components" and the PMF component names were changed to: *"sea salt, mineral dust, ammonium neutralised, fugitive dust,* and *industry*."

**Line 499: I'd see this discussion better located in section 4.2. with also the quantification of the resulting PM mass**

This discussion was merged into one section on source apportionment as suggested, thank you.

**Line 552: the "source profile" has nothing to do with the fraction of the PM accounted by a specific source. The word "profile" should be removed here and, as stated before, I'd prefer "component" instead of "source"**

Considering outputs of the PMF model as mass fractions of the total, "components" is a better descriptor and has been corrected throughout the document. Thank you.

**Line 570: I find not surprising that sea salt is present in all the sources detected in a coastal site.**

We agree. This is now added at lines 309-320 "An Fpeak strength of 0.5 was used to retain the best PMF solution whose five components (sea salt, mineral dust, ammonium neutralised, fugitive dust and industry) are shown in Figure 6. The relative contribution of those components to the total estimated mass is shown in Figure S3. Sea salt accounted for the largest fraction of the (mass concentration (: 74.7 ± 1.9%). Mineral dust accounted for (15.7 (± 1.4%,) of the evaluated total mass concentration. The remaining fraction was accounted by three components characterised by secondary species and heavy metals, ammonium neutralised (6.1 ± 0.7%), fugitive dust (2.6 ± 0.2%) and industry (0.9 ± 0.7%). However, the major tracers of the sea salt component, Na+ and Cl-, were ubiquitous in all components, not surprising considering the continuous inflow of marine air to HBAO. As it can be seen in Figure 6, Na+ and Cl- contributed to 35.2 ± 5.8% of their mass to the mineral dust component. to 47.4 (± 1.9%) of the mass of the fugitive dust PMF component 1.3 (± 17.8%) of the mass of the industry component".

**Figure 6: the fits in the bottom panels are misleading: there is no correlation between the plotted variables and the equations (by the way: the uncertainty on slope and bias should be given) have no sense.**

You are correct, considering that the axis scales differ, the trend lines appeared misleading. The axes were changed and the trendlines removed from Figure 6 (now Figure 7).

**In addition I have a more general comment: in my opinion the importance of the article is in the large and detailed set of data that have been collected.  The PMF analysis does not add so-much (and I see that in the conclusions its results poorly commented) since the quantification of the impact (fraction of the PM mass) of each "source" (I do not like the term as it has been used int he text, see comments in the PDF) is not very firm due to the sea salt "contamination "basically in all the detected factors. I'd invite the Authors to add in the text some more comments on the significance of the PMF exercise.**

We agree, see general comments.

**Chemical composition and source apportionment of atmospheric aerosols on the Namibian coast**

**Danitza Klopper[1], Paola Formenti[2], Andreas Namwoonde[3], Mathieu Cazaunau[2], Servanne Chevaillier[2], Anaïs Feron[2], Cécile Gaimoz[2], Patrick Hease[2], Fadi Lahmidi[2], Cécile Mirande-Bret[2], Sylvain Triquet[2], Zirui Zeng[2] and Stuart J. Piketh[1]**

[1] North-West University, School for Geo- and Spatial Sciences, Potchefstroom, South Africa

[2] Laboratoire Interuniversitaire des Systèmes Atmosphériques (LISA), UMR CNRS 7583, Université Paris-Est-Créteil, Université de Paris, Institut Pierre Simon Laplace, Créteil, France

[3] SANUMARC, University of Namibia, Henties Bay, Namibia

**Corresponding author:** paola.formenti@lisa.ipsl.fr

**Abstract**

The chemical composition of aerosols is of particular importance to assess their interactions with radiation, clouds and trace gases in the atmosphere, and consequently their effects on air quality and the regional climate. In this study, we present the results of the first long-term dataset of the aerosol chemical composition at an observatory on the coast of Namibia, facing the southeast Atlantic Ocean. Aerosol samples in the mass fraction of particles smaller than 10 μm in aerodynamic diameter ($PM_{10}$) were collected during 26 weeks between 2016 and 2017 at the ground-based Henties Bay Aerosol Observatory (HBAO; 22°6'S, 14°30'E, 30 m above mean sea level). The resulting 385 filter samples were analysed by X-ray fluorescence and ion-chromatography for 24 inorganic elements and 15 water-soluble ions.

Source apportionement was conducted by looking at the inter-elemental and ionic c

Statistical analysis by Positive Matrix Factorisation (PMF) identified five major components, *sea salt* (mass concentration: 74.7 ± 1.9%), *mineral dust* (15.7 ± 1.4%), *ammonium neutralised* (6.1 ± 0.7%), *fugitive dust* (2.6 ± 0.2%) and *industry* (0.9 ± 0.7%). While the contribution of sea salt aerosol was persistent, as the dominant wind direction was south-westerly and westerly from the open ocean, the occurrence of mineral dust was episodic and coincided with high wind speeds from the south-southeast and the north-northwest, along the coastline. Concentrations of heavy metals measured at HBAO were higher than reported in the literature from measurements over the open ocean. V, Cd, Pb and Nd were attributed to fugitive dust emitted from bare surfaces or mining activities. As, Zn, Cu, Ni and Sr were attributed to the combustion of heavy oils in commercial ship traffic across the Cape of Good Hope sea route, power generation, smelting and other industrial activities in the greater region. Fluoride concentrations up to 25 µg m$^{-3}$ were measured, as in heavily polluted areas in China. This is surprising and a worrisome result that has profound health implications and deserves further investigation. Although no clear signature for biomass burning could be determined, the PMF *ammonium neutralised component* was described by a mixture of aerosols typically emitted by biomass burning, but also by other biogenic activities. Episodic contributions with moderate correlations between NO$_3^-$, nss-SO$_4^{2-}$ (higher than 2 µg m$^{-3}$) and nss-K$^+$, were observed, further indicative of the potential for an episodic source of biomass burning.

Sea salt accounted for up to 57% of the measured mass concentrations of SO$_4^{2-}$ and the non-sea salt fraction was contributed mainly by the *ammonium neutralised* component, and small contributions from the *mineral dust* component. The marine biogenic contribution to the *ammonium neutralised* component 
[revised manuscript text omitted]
 evaluated as 10%. This represents the upper limit uncertainty, taking into account:

- The uncertainty related to the uniformity of the aerosol deposit on the filters, and the scaling error that can occur due to the fact that the area of the deposit which is analysed is smaller than the area of the aerosol deposit;
- The statistical error on the photon counts, in particular for trace elements whose concentrations are close to their detection limits;
- The percent error on the certified mono- and bi-elemental standard concentrations (Micromatter Inc., Surrey, Canada) used for calibration of the XRF apparatus;
- For the lightest elements (Z < 20, Na to Ca), the choice of the correction factor to account for the self-attenuation of the X-ray signal, in particular for particles larger than 1 µm in diameter (Formenti et al., 2010). Constant correction factors (Table S1) were estimated through the sampling period assuming a mean diameter of 4.5 μm to represent the average coarse particle size.

[revised manuscript text omitted]

- Heavy-oil combustion from industry and commercial shipping as well as mining activities traced by elements such as Ni, V, Pb, Cu, Zn (Ettler et al., 2011; Becagli et al., 2017; Johansson et al., 2017; Kříbek et al., 2018; Sinha et al., 2003; Soto-Viruet, 2015; Vouk and Piver, 1983), and;

- Seasonal transport of biomass burning aerosols traced by nss-K$^+$ (Andreae et al., 1998; Andreae and Merlet, 2001). Nss-K+ was calculated from measured K$^+$ assuming the mass ratio K$^+$/Na$^+$ of 0.036 as in seawater (Seinfeld and Pandis, 2006).

**3.2    Positive matrix factorisation**

Multivariate statistical methods such as positive matrix factorisation (PMF) are widely used to identify components or 'source' profiles and explore source–receptor relationships using the trace element compositions of atmospheric aerosols (e.g., Schembari et al., 2014; Hopke and Jaffe, 2020). The PMF uses weighted least-squares component analysis to deconvolute the matrix of observed values (X) as $X = G$ x $F + E$, where $G$ and $F$ are the matrices representing the component scores and component loadings, respectively, and $E$ is the matrix of residuals equal to the difference between observed and predicted values (Paatero and Tapper, 1994; Paatero et al., 2014).

In this paper, the multivariate PMF statistical analysis was conducted with the EPA (Environmental Protection Agency) PMF version 5.0 (Norris et al., 2014).  The XRF and IC datasets were combined by retaining only elements/ions measured above the MQL in more than 70 samples (that is, at least in 20%

of the collected values). This criterion excluded Ba, Br⁻, PO₄⁻ and Mn²⁺. Occasional missing values in the retained elements/ions were replaced by the species median value, as recommended by Norris et al. (2014). Uncertainties for missing values were replaced by a dummy value (99999) to ensure that these samples do not skew the model fit (Norris et al., 2014). In order to weight the concentrations according to their amount, a relative uncertainty of 10%, 20% and 60% was attributed to each value of concentration in the input matrix based on their ratio to their respective MQL (larger than 3.3, comprised between 1.25 and 3.3, and comprised between 1 and 1.25, respectively). The final input matrix comprised 385 observations of 33 chemical species. The water-soluble ionic form instead of the elemental form was retained for Mg, Na, Cl, K, Ca and S.

Based on the temporal correlation, the PMF analysis resolves the chemical dataset into a user-specified number of components ('sources'). No completely objective criterion exists for selecting the number of components and so the model was run considering potential solutions of three to seven sources. Each of these models were run 100 times using randomised seeds. For each of these runs, the robustness-of-fit was compared and the estimation of the error range of each solution was done by running a classical bootstrap analysis, displacing chemical species in each modelled component and testing the rotational ambiguity of the solutions, and finally also by running a supplementary bootstrap analysis enhanced by displacement of component elements (Norris et al., 2014; Paatero et al., 2014). *Fpeak* rotations with strengths between -0.5 and 1.5 were tested to further optimise the component solutions.

**4. Results and discussion**

**4.1 Meteorological conditions during sampling**

[revised manuscript text omitted]

5.

An *Fpeak* strength of 0.5 was used to retain the best PMF solution whose five components (*sea salt,*

*mineral dust, ammonium neutralised, fugitive dust and industry)* are shown in Figure 6. The relative contribution of those components to the total estimated mass is shown in Figure S3. *Sea salt* accounted for the largest fraction of the (mass concentration (74.7 ± 1.9%). Mineral dust accounted for (15.7 (±

1.4%) of the evaluated total mass concentration. The remaining fraction was accounted by three components characterised by secondary species and heavy metals, *ammonium neutralised* (6.1 ± 0.7%),

*fugitive dust* (2.6 ± 0.2%) and *industry* (0.9 ± 0.7%). However, the major tracers of the *sea salt*

component, Na⁺ and Cl⁻, were ubiquitous in all components,  not surprising considering the continuous inflow of marine air to HBAO. As it can be seen in Figure 6, Na⁺ and Cl⁻  contributed to

35.2 ± 5.8% of their mass to the mineral dust component. to 47.4 (± 1.9%) of the mass of the *fugitive*

*dust* component, and to 1.3 (± 17.8%) of the mass of the *industry* component)

**4.2.1 Sea salt**

As expected, the major tracers of sea salt aerosols (Cl⁻, Na⁺, Mg²⁺ and K⁺) were sampled in high concentrations (up to 76, 53, 5.6 and 2.0 µg m⁻³, respectively) throughout the sampling periods. Their time variability, illustrated in Figure 5 by the example of Na⁺, was very similar and characterised by a significant continuous background that could be represented by a 10-point moving average (that is, 90

hours). The calculated mean background concentration was 10.1 ± 3.6 µg m⁻³. No seasonal cycle was evident due to the dominance of southerly and south-westerly winds transporting marine air masses onshore (Figure 3).

The PMF *sea salt* component was represented by Na⁺, Cl⁻, Mg²⁺, K⁺, Ca²⁺ and SO₄² (Figure 6), and accounted for 74.7 ± 1.9% of the total aerosol mass (Figure S3). Table 2 shows the mass ratios of Cl⁻,

Mg²⁺, K⁺, Ca²⁺, F⁻ and SO₄²⁻ to Na⁺ for 2016 and 2017, calculated as the slopes of their linear regression lines, and evaluated by the coefficient of determination (*R²*). This table also gives the slope of the linear regression lines for the PMF *mineral dust* component. The experimental values were compared with average ratios in seawater (Seinfeld and Pandis, 2006). The average Cl⁻/Na⁺ mass ratio was 1.4 ± 0.1 in

2016 and 1.3 ± 0.1 in 2017 (also consistent for the PMF *sea salt* component), lower by 25% than the value expected in seawater of 1.8. This difference has previously been reported in fresh sea salt in acidic marine environments (e.g., Zhang et al., 2010), and is attributed to $Cl^-$ depletion via reactions between

NaCl and sulfuric- and nitric acids. A very good correlation was observed between the ratios of $Mg^{2+}$ (0.12

± 0.01) and $K^+$ (0.04 ± 0.01) to $Na^+$ in this data set and the value reported for sea water (Table 2) (Seinfeld and Pandis, 2006). Conversely, the linear correlation between $Ca^{2+}$ and $Na^+$ (not shown) was less pronounced ($R^2$ = 0.61 and 0.42 in 2016 and 2017, respectively). The $Ca^{2+}/Na^+$ mass ratio was systematically higher than in seawater (0.04), indicating the contribution of crustal calcium typical of the Namibian soils (see section 4.2.2).

Using the average seawater ratio, the mean sea-salt (ss) $Ca^{2+}$ concentration was estimated as 470 ± 360

ng $m^{-3}$ and 360 ± 210 ng $m^{-3}$ for 2016 and 2017, respectively. The mean non-sea-salt (nss) $Ca^{2+}$

concentration was 420 ± 520 and 270 ± 400 ng m-3, respectively for the two years, representing 47%

and 42% of the mean measured $Ca^{2+}$ concentrations. Similarly, for both 2016 and 2017, the ss and nss components of $K^+$ were estimated as 367 ± 246 ng $m^{-3}$ and 44 ± 54 ng $m^{-3}$ respectively, accounting for

89% and 11% of the $K^+$ mass. The PMF estimated that *sea salt* contributed to 53.0 ± 1.6% of the calcium and 75.1 ± 2.4% of the $K^+$ mass.

The mean $F^-/Na^+$ mass ratio measured at HBAO was 0.39 ± 0.29 in 2016 and 0.32 ± 0.29 in 2017 and was 0.19 ± 0.01 for the PMF *sea salt* component, enriched by two to four orders of magnitude to the average seawater composition (mass ratio 1.2 x $10^{-4}$; Table 2).

**4.2.2 Mineral dust**

The PMF *mineral dust* component, composed by Si, Al, Fe, Ti, $Ca^{2+}$, Mn, P, $F^-$ and V (Figure 6), accounted for 15.7 ± 1.4% of the total estimated mass. The time series of Al and nss-$Ca^{2+}$ (Figure 5) were analysed to investigate the temporal variability of airborne mineral dust at Henties Bay. The mean concentrations of mineral dust elements Al, Fe, Ti and Si were higher for night-time sampling between 21 and 06 UTC, and lower in the day (9 to 18 UTC), in correspondence of easterly winds which were only observed at night and in the early morning (Figure 4).

Differently from sea salt, the occurrence of mineral dust was not continuous but episodic. Episodes of mineral dust corresponded to times when the concentrations of Al and nss-$Ca^{2+}$ exceeded background values (modelled as the 10-point moving average) for a minimum of 3 consecutively sampled filters.

Similar time variability was observed for elemental Fe, Si, Ti and P (not shown). Overall, 19 episodes of mineral dust were identified during the two years of sampling (Table S2).

The mean mass concentration of elemental Al was 556 ± 643 ng $m^{-3}$ in 2016 and 446 ± 551 ng $m^{-3}$ in

2017, while values peak as high as 4.7 µg $m^{-3}$ (Table 1). To the best of our knowledge, no other measurements of Al are available in Namibia for comparison. Our arid sampling site is surrounded by loose sand, gravel plains (Matengu et al., 2019) and the deep Omaruru river valley directly north of the sampling site which is also a recognised source of mineral dust to the offshore waters (Tlhalerwa et al.

2012). While mostly characterised by gravels, some clay-rich deposits are found around the river valley approximately 17 km northeast of HBAO (Matengu et al., 2019). The relatively low aluminium concentrations measured at HBAO suggest that these are not a major local source for the site. The nss-

$Ca^{2+}$ annual mean at HBAO (703 ± 644 ng m$^{-3}$ in 2016 and 428 ± 437 ng m$^{-3}$ in 2017) is similar to the concentrations (mean 425 ng m$^{-3}$ and maximum of 800 ng m$^{-3}$) measured in central Namibia at Gobabeb, in the Namib Desert (23°45'S, 15°03'E; Annegarn et al., 1983). This is also the case for Fe, who annual mean concentrations at HBAO (372 ± 480 ng m$^{-3}$ in 2016 and 338 ± 433 ng m$^{-3}$ in 2017) compare well with the average of 246 ng m$^{-3}$ (Annegarn et al., 1983).

Table 3 shows the mass ratios for major components of mineral dust as well as some heavy metals (V

and Ni). Overall, Si, Fe, and Ti showed very good correlations to Al as expected for mineral dust ($R^2$ >

0.9). The average mass ratio of Si/Al was 3.7 ± 1.0 in 2016 and 3.4 ± 0.8 in 2017, lower than the average values of 4 to 4.6 expected in global soils and crustal rock (Seinfeld and Pandis, 2006). This is attributed to the size-fractionation during aeolian erosion of soils producing airborne dust. As a matter of fact, our average values are consistent with those obtained for particles less than 10 μm in diameter by Eltayeb et al. (1993) at Gobabeb. Our averages, generally higher than in mineral dust from north Africa (Formenti et al., 2014), compare well with the value (3.4) reported by Caponi et al. (2017) for mineral dust aerosols generated in a laboratory experiment from a soil collected to the northeast of HBAO. The average Fe/Al ratio was 0.74 ± 0.19 in 2016 and 0.76 ± 0.18 in 2017 (0.8 ± 0.3 for the PMF solution), lower than the ratio of 1 reported by Eltayeb et al. (1993). The same is observed for the Ti/Al ratio, which was 0.07 ± 0.22 in 2016 and 0.06 ± 0.03 in 2017 (0.08 ± 0.01 in the PMF solution), while approximately 0.15 in Eltayeb et al. (1993).

The average nss-$Ca^{2+}$/Al ratio was 1.3 ± 0.7 in 2016 and 1.4 ± 0.7 in 2017, however, for the strongest dust episodes (Al values higher than 1 μg m$^{-3}$) the ratio tended to 1 (Figure 7). This is in agreement with the specific mineralogy of Namibian soils that are rich in limestone and gypsum (Annegarn et al., 1983;

Eltayeb et al., 1993). The PMF analysis attributed 40.5 ± 0.6% of the total $Ca^{2+}$ mass to the *mineral dust*

*component,* of the same order of magnitude than obtained from the chemical apportionement (nss fraction representing 47% of the total -$Ca^{2+}$). The $SO_4^{2-}$/$Ca^{2+}$ mass ratio in the PMF *mineral dust* was 1.1

± 0.2, three to four times lower than that the nss-$SO_4^{2-}$/nss-$Ca^{2+}$ obtained from chemical apportionment and about half the mass ratio for gypsum, which, however, well coincided with the mass ratio obtained for when selecting the dust episodes only. The mean Fe/nss-$Ca^{2+}$ ratio was 0.54 ± 0.23 in 2016 and 0.65

± 0.23 in 2017, higher than the value of 0.11 ± 0.10 reported by Caponi et al. (2017), pointing to the diversity in soil mineralogy, even at relatively small spatial scales.

As for nss-Ca$^{2+}$, values for nss-K$^+$/Al ratios (Figure 7) were spread but ranged between 0.1 and 0.5 when Al concentrations exceeded 1 µg m$^{-3}$. These values are in agreement with those for mineral dust sources in North Africa (Formenti et al., 2014). The PMF K$^+$/Al mass ratio was 0.16 ± 0.01, in good agreement with the average nss-K$^+$/Al (0.13 ± 0.12) by chemical apportionment and half of that reported in the literature (0.25 – 0.45, Eltayeb et al., 1993).

The average phosphorus concentrations measured at HBAO were 11 ± 9 ng m$^{-3}$ in 2016 and 14 ± 4 ng m$^{-3}$ in 2017. Phosphorous was very well correlated with Al in 2016 ($R^2$ = 0.92) and only moderately correlated in 2017 ($R^2$ = 0.66). The P/Al mass ratio annual average was 0.03 ± 0.02 in 2016 and 0.05 ± 0.02 in 2017 (0.01 ± 0.01 in the PMF *mineral dust*). As was observed for the nss-Ca$^{2+}$/Al, the P/Al ratio tended to an asymptotic value of 0.02 when Al exceeded 1 µg m$^{-3}$ (not shown). The PMF result is closer to that reported by Formenti et al. (2003a) for the outflow of Saharan dust to the North Atlantic Ocean (0.0070 ± 0.0004).

**4.2.3  Heavy metals**

The PMF identified two components characterised by heavy metals: a *fugitive dust* component (traced by V, Cd, Pb, Nd and Sr) and an *industry* component, characterised by As, Zn, Cu, Ni and Sr, representing 2.6 (± 0.2%) and 0.9 (± 0.7%) of the total estimated mass.

Vanadium and nickel are naturally occurring in mineral deposits in soils (Annegarn et al., 1983; Maier et al., 2013), but they are also known tracers of heavy-oil combustion, as reported in Becagli et al. (2013) and references therein. Their average concentrations at HBAO were 9 ± 5 ng m$^{-3}$ (2016) and 7 ± 6 ng m$^{-3}$ (2017) for V, and 8 ± 7 ng m$^{-3}$ (2016) and 7 ± 4 ng m$^{-3}$ (2017) for Ni. The highest V concentrations corresponded to south-south-easterly winds while high Ni concentrations were measured in the south-west wind sector (Figure S4). The annual mean values of V and Ni at HBAO are an order of magnitude larger than measured over the open ocean by Chance et al. (2015), higher than those reported by Hedberg et al. (2005) at towns affected by copper smelters and comparable to those measured by Isakson et al. (2001) at a Swedish harbour and by Becagli et al. (2012) in the central Mediterranean Sea downwind of a major shipping route.

Vanadium was well correlated with Al when Al exceeded 1 µg m$^{-3}$ ($R^2$ around 0.4), whereas no correlation between Ni and Al were observed (Figure 7). Additionally, the correlation of V to Si, also used in the literature as a tracer of mineral dust, was evident while moderate ($R^2$ around 0.4), and no correlation was found for Ni. This differs from what was reported by Becagli et al. (2012), who found that neither V nor Ni were correlated to Si. In our dataset and the PMF *mineral dust* component (section 4.2.2), both V/Si and Ni/Si ratios were enriched by a factor of 10 or more to reference values for the upper continental crust (3.1×10$^{-4}$ and 1.5×10$^{-4}$ for V/Si and Ni/Si, respectively; Henderson and Henderson, 2009). The V/Ni mass ratio was 1.7 ± 1.1 for 2016 and 1.3 ± 1.3 in 2017, lower than reported by Lyyränen et al. (1999) and Corbin et al. (2018) for heavy fuel oil in diesel engines, and by Becagli et al. (2017) and Viana et al. (2009) in the Mediterranean basin ambient air (2.8–2.9 and 4–5, respectively).

All these elements, and furthermore their poor correlation ($R^2$ around 0.3), suggest that V and Ni do not necessarily have the same sources. Mining activities, likely in the Otavi mountain area (Boni et al., 2007), should account for the high concentrations of V, with additional contributions from heavy-oil combustion, where V is present as an impurity (Isakson et al., 2001, and references therein; Vouk and Piver, 1983). On the contrary, combustion of heavy-oils seems to be the primary source of Ni.

This hypothesis is supported by the PMF analysis. The PMF apportionnement of V and Ni concentrations (Figure S5) clearly distinguishes the relative source contributions, and preferentially associates V with *mineral dust* and *fugitive dust* components, while Ni with the industry component.

Moderate to good correlations of V and Ni with Zn ($R^2$ of 0.42 and 0.55, respectively), Cu (0.55 and 0.73) and Pb (0.56 and 0.69) were observed in the dataset. Zn and Pb are found as impurities in bulk fuels for ships (Isakson et al., 2001) and also from copper smelting, as reported in central Chile (Hedberg et al. 2005) urban air in the United States of America (Ramadan et al., 2000). The mean concentration of Zn at HBAO (11 ± 9 ng m$^{-3}$) was about two orders of magnitude higher than over the southeast Atlantic Ocean (Chance et al., 2015) and in air over the arid landscapes (Annegarn et al., 1983). Likewise, the mean Pb concentration (75 ± 89 ng m$^{-3}$) was three orders of magnitude higher than reported by Chance et al. (2015) for soluble Pb and comparable to values measured in the western Mediterranean by Denjean et al. (2016). The PMF seperates the largest fractions of Zn and Pb into the *industry* and *fugitive dust* components, respectively. Although some of these heavy metals may be sourced from the commercial shipping route offshore, the mass ratios for tracer elements were not in agreement with our results and so we cannot conclusively state shipping heavy-oil combustion as the source of these heavy metals.

Average concentrations of Cu at HBAO were 8 ± 6 ng m$^{-3}$, an order of magnitude higher than measured in windblown dust by Annegarn et al. (1983) in the central Namib but two orders of magnitude smaller than the average measured by Lee et al. (1999) in highly polluted Hong Kong (125.1 ng m$^{-3}$). Ettler et al. (2011) showed that copper ore mining and smelting operations in the Zambian Copperbelt are a significant source of potentially bioavailable copper, that, unlike phosphorus, has been found to inhibit plankton growth in laboratory studies (Paytan et al., 2009) and over the western Mediterranean (Jordi et al., 2012). Similar contamination of topsoil was found by Kříbek et al. (2018) at operations in the Tsumeb mining district, Namibia (19°14'S, 17°43'E). Average Cu concentrations were comparable to values 4.9 ± 11.5 ng m$^{-3}$ reported for a town closer to smelters in Chile, and an order of magnitude smaller than in the urban environement of the the capital city of Santiago (77.5 ± 78.2 ng m$^{-3}$; Hedberg et al., 2005). The Cu/Ni ratio (1.24 ± 0.20) in the PMF *fugitive dust* component was about half than reported for soil samples polluted by copper mine tailings from the Gruben River valley (2.03 ± 2.30,
Taylor and Kesterton, 2001).

The mean mass concentration of Cd was 1502 ± 1458 ng m$^{-3}$ in 2016 and 219 ± 163 ng m$^{-3}$ in 2017. The
difference is mainly owing to high concentrations in October of 2016 which coincided with high
concentrations in all other heavy metals, except for As. Cd concentrations in 2016 were less than that
reported for airborne road dust (7.4 ± 7.8 µg m$^{-3}$) and our 2017 concentrations were in the order of that
measured in ambient air (0.14 ± 0.04 µg m$^{-3}$) in the seaside city of Khobar, Saudi Arabia (El-Sergany and
El-Sharkawy, 2011). The Cd/Pb ratio of 9.96 ± 0.21 for the PMF *fugitive dust* component was slightly
higher than 7.14 ± 4.26 in the ambient air of the coastal desert environment in Khobar (El-Sergany and
El-Sharkawy, 2011). The correlation of Pb, Nd and Sr the *fugitive dust* component may indicate
contributions of non-micaceous kimberlites from a variety of source regions across southern Africa
(Smith, 2001). The Sr/Nd ratio for the *fugitive dust* component (3.58) was close to 3.35 reported for
Kimberlites at Uintjiesberg in the Northern Cape of South Africa.

4.2.34.2.4    **Fluoride**

One of the striking features of Table 1 is the high mean concentration of F$^-$ measured at HBAO (4.3 ± 4.0
µg m$^{-3}$ in 2016 and 2.8 ± 2.5 µg m$^{-3}$ in 2017), with peak values as high as 25 µg m$^{-3}$. Those annual mean
concentrations were comparable to the mean 24-h fluoride concentrations measured between 1985 and
1990 over the South African Highveld by Scheifinger and Held (1997). The measured concentrations at
HBAO were also comparable to those of heavily polluted areas in China (Feng et al., 2003), and
significantly higher than reported for Europe, even in the polluted Venice lagoon (Prodi et al., 2009) or
in areas nearby ceramic and glass factories (Calastrini et al., 1998). The peak values at HBAO were
significantly higher than maxima reported by these authors and ranging between 1.4 and 2.9 µg m$^{-3}$. The
highest F$^-$ concentrations were associated with south to easterly winds, that is, from the subcontinent
(not shown). The very good correlation of F$^-$ with nss-Ca$^{2+}$, shown in Figure S6 (*R$^2$* equal to 0.76 in 2016
and to 0.84 in 2017), yielded a mean mass ratio of 6.4 and 5.8, respectively, much higher than reported
in groundwater, aerosols or precipitation in polluted environments (Feng et al., 2003; Prodi et al., 2009).

The strong relationship to nss-Ca$^{2+}$ (and *a posteriori* to Ca$^{2+}$) drove the PMF apportionment (Figure S7),
which attributed approximately the 94% of the F$^-$ mass concentrations to the sea salt and mineral dust
components (55.1 ± 1.9% and 38.8 ± 1.1%, respectively), and the remaining 6% to *fugitive dust* (2.3 ±
0.5%) and *industry* (3.8 ± 1.0%). Possible sources are the emission of fugitive dust during fluorspar
mining of carbonatite related fluorospar deposits at the Okorusu Mine (20°3'S, 16°44'E), but very likely
also from the periodic surface mining occurring approximately 20 km south of HBAO to provide gravel
for the construction of a major road between Swakopmund and Henties Bay which started late in 2015
(A. Namwoonde, *pers. corr.*). The evaporation of fluoride rich water, leached into groundwater (Wanke
et al., 2015, 2017) from fluoride-rich mineral deposits and soils, throughout the region and in the coastal waters (Compton and Bergh, 2016; Mänd et al., 2018), would also increase atmospheric F- concentrations. In an analysis of borehole water in Namibia, roughly 80% of those sites surveyed were deemed unsafe to drink as a direct result of high fluoride concentrations (Christelis and Struckmeier, 2011).

**4.2.5 Arsenic**

The annual mean of the arsenic concentrations at HBAO was $22 \pm 16$ ng m$^{-3}$ in 2016 and $239 \pm 344$ ng m$^{-3}$ in 2017. The mean for 2017 is skewed due to two sampling weeks with very high concentrations in the order of those measured in rural and urban-industrial areas affected by mining and smelting emission sources (Hedberg et al., 2005; Šerbula et al., 2010).

The PMF analysis exclusively associated As the *industry* component along with large fractions of the Zn, Cu, Ni, Sr and Co. Known sources of atmospheric of arsenic are biomass burning, heavy-oil combustion and non-ferrous metal smelting operations (Ahoulé et al., 2015; Gomez-Caminero et al., 2001). A possible local source could be the Tsumeb smelter in the northeast of HBAO (KPMG, 2014).

The PMF As/Zn, As/Pb and Zn/Pb ratios were $9.0 \pm 0.3$, $6.4 \pm 0.8$ and $0.7 \pm 0.1$, in good agreement with those reported by Hedberg et al. (2005) for a copper smelter plume in Chile ( 7.7, 4.5 and 0.6, respectively). This is in good agreement with the fact that no correlations between As to Al nor nss-Ca$^{2+}$ were found, ruling out any major contribution of Inorganic arsenic in geologic formations released from mining operations or evaporated from soil and groundwater (Gomez-Caminero et al., 2001). Likewise, no discernible correlation between As and MSA was found, suggesting only a minor release of arsenic by marine algae and plankton (Sanders and Windom, 1980; Shibata et al., 1996).

**4.2.4 4.2.6    Secondary aerosols and sulphate**

The PMF *ammonium neutralised* (Figure 6) comprised secondary species such as by SO$_4^{2-}$, NH$_4^+$, MSA, oxalate, and nitrate, which accounted for $6.1 \pm 0.7\%$ of the estimated aerosol mass.

The annual mean sulphate concentration measured at HBAO was $4.1 \pm 2.6$ µg m$^{-3}$ in 2016 and $3.4 \pm 1.4$ µg m$^{-3}$ in 2017 (Table 4), higher than previously measured over the southern Atlantic and Pacific oceans (Zhang et al., 2010) and comparable to springtime measurements in the Venice Lagoon (Prodi et al., 2009). As already discussed in Formenti et al. (2019), the highest concentrations were measured in spring and autumn, while minima occurred between May and August. SO$_4^{2-}$ and Na$^+$ showed good correlation ($R^2 = 0.92$ in 2016 and 0.83 in 2017, Table 2). However, their annual mass ratios ($0.36 \pm 0.14$ and $0.42 \pm 0.23$ in 2016 and 2017, respectively) were higher than the expected mass ratio in seawater (0.25; Seinfeld and Pandis, 2006), which was used as nominal reference to apportion SO$_4^{2-}$ into its ss and nss fractions. As a result, up to 57% of the measured SO$_4^{2-}$ mass concentration in the PM$_{10}$ fraction was attributed to sea salt aerosols, while the nss-component was of the order of 43%. The PMF estimated that the *sea salt* component contributed to 66.6 ± 0.4% of the total sulphate mass. This is in agreement with previous observations in the south Atlantic Ocean (Andreae et al., 1995; Zhang et al., 2010; Zorn et al., 2008). On the contrary, at the remote Brand se Baai site along the Atlantic coast of South Africa (31.5°S, 18°E), Formenti et al. (1999) reported that sea salt accounted for about 92% of the total measured elemental sulphur concentrations.

The MSA concentrations measured at the site ranged between 10 and 230 ng m$^{-3}$ (Table 1). The mean annual concentration was 63 ± 39 ng m$^{-3}$, three times higher than the mean value of 20 ± 20 ng m$^{-3}$ (6.2 ± 4.2 ppt) reported by Andreae et al. (1995) over the open ocean along 19°S, and lower than in the southeast Atlantic Ocean (Zhang et al., 2010; Table 4). As already described in Formenti et al. (2019), the MSA concentrations were higher in the austral summer and spring and lower in the austral winter. DMS is more efficiently oxidised in warmer conditions (Ayers et al., 1997; Huang et al., 2017) which explains the the higher daytime mean concentrations of marine biogenic products (MSA and nss-SO$_4^{2-}$) and lower means at night and in the winter. Springtime averages for MSA were in the range of that measured by Huang et al. (2018) during a springtime cruise over the South Atlantic and by Prodi et al. (2009) in the Venice Lagoon (Table 4). The mismatch of seasonality with respect to that of the phytoplankton blooms (Louw et al., 2016) is already discussed by Formenti et al. (2019) and attributed to the spread of blooms in the BUS region depending on local conditions.

[revised manuscript text omitted]

While the coarse resolution of air mass backtrajectories and the dominance of marine air masses does not allow to distinguish sources at the country scale, the PMF analysis performed in this paper was able to identify the specific and distinct contribution of mining activities, including for road construction for the majority of the heavy metals (ex., V). Our results shown that mining activities severely affect the air quality and contribute to concentrations as high as, or even higher than in well-known polluted regions of the world, such as the Venice lagoon (Prodi et al., 2009). The persistence of these high concentrations over the two years of sampling is extremely worrying for the affected populations and needs to be addressed by dedicated investigations and decision-making procedures. We suspect that some of that contamination, contributing to the highest heavy metal concentrations in October 2016, might be due to fugitive dust released by the major road construction between Walvis Bay, past Henties Bay and towards Angola, that started in the second half of 2016. Having said this, that specific week discarded, there is no significant difference between the concentration levels in 2016 (before road works) and

2017 (during the road works), suggesting that the pollution by heavy metals is a specific feature in the region, with likely implications on weather and climate. One such effect could be the deposition of these metals in the ocean. The deposition of macronutrients (P, Fe) from the outflow of mineral dust is not expected to be relevant for the BUS region, one of the most productive marine environments in the world, while it could be important in fertilising waters near the coast (Dansie et al., 2017) and in the Southern Ocean (Okin et al., 2011). On the other hand, the atmospheric deposition of trace metals (Cr, Cu, Ni, Mn, or Zn) in the aerosols, which play a biological role in enzymes and as structural elements in proteins (Morel and Price, 2003), could affect the marine productivity of the BUS and should be explored in future work. The complexity and diversity of sources that might contribute to the aerosol population at HBAO , deserve further dedicated investigation.

[revised manuscript text omitted]

Christelis, G. and Struckmeier, W. (eds.): Groundwater in Namibia, an explanation to the Hydrogeological Map, 2nd ed. Department of Water Affairs, Division Geohydrology, 2011. Available from: https://www.deutsche-rohstoffagentur.de/EN/Themen/Wasser/Projekte/abgeschlossen/TZ/Namibia/groundwater_namibia.pdf?__blob=publicationFile&v=3 Date of access: 5 Aug. 2020.

Cole J. and Villacastin C. Sea surface temperature variability in the northern Benguela upwelling system, and implications for fisheries research, Inter. J. Rem. Sens., 21, 1597-1617, https://doi.org/10.1080/014311600209922, 2000.

Compton, J.S. and Bergh E.W.: Phosphorite deposits on the Namibian shelf, Marine Geology, 380, 290-314. https://doi.org/10.1016/j.margeo.2016.04.006, 2016.

[revised manuscript text omitted]

Gillett, R., Galbally, I., Ayers, G., Selleck, P., Powell, J., Meyer, M., Keywood, M. and Fedele, R.: Oxalic acid and oxalate in the atmosphere. In Proceedings of the 4th IUAPPA World Congress Clean Air Partnerships: Coming Together for Clean Air, Brisbane, Australia, 9–13 September 2007. Available online: http://www.cmar.csiro.au/e-print/internal/2007/gillettrw_xa.pdf

Gomez-Caminero, A., Howe, P., Hughes, M., Kenyon, E., Lewis, D., Moore, M., Ng, J., Aitio, A. and Becking, G.: Arsenic and Arsenic Compounds, Environmental Health Criteria no. 224, 2nd ed. Geneva:World Health Organization, 2001.

Hedberg, E., L. Gidhagen, and C. Johansson, C., Source contributions to PM10 and arsenic concentrations in Central Chile using positive matrix factorization, Atmos. Environ., 39, 549–561, https://doi.org/10.1016/j.atmosenv.2004.11.001, 2005.

Heine, K. and J. Völkel, Soil Clay Minerals in Namibia and Their Significance for the Terrestrial and Marine Past Global Change Research, Afr. Study Monogr., 40, 31–50, https://doi.org/10.14989/96299, 2010.

Henderson, P. and Henderson, G.M. (Eds.): The Cambridge Handbook of Earth science data, Cambridge University Press, London, 2009.

Hopke, P.K. and Jaffe, D.A.: Letter to the Editor: Ending the Use of Obsolete Data Analysis Methods,
Aerosol Air Qual. Res., 20, 688-689, https://doi.org/10.4209/aaqr.2020.01.0001, 2020.

Huang, S., Poulain, L., Van Pinxteren, D., Van Pinxteren, M., Wu, Z., Herrmann, H. and Wiedensohler, A.:
Latitudinal and Seasonal Distribution of Particulate MSA over the Atlantic using a Validated
Quantification Method with HR-ToF-AMS, Environ. Sci. Technol., 51, 418–426,
https://doi.org/10.1021/acs.est.6b03186, 2017.

Isakson, J., Persson, T.A. and Lindgren, E.S.: Identification and assessment of ship emissions and their
effects in the harbour of Göteborg, Sweden, Atmos. Environ., 35, 3659–3666,
https://doi.org/10.1016/S1352-2310(00)00528-8, 2001.

Johansson, L., Jalkanen, J. and Kukkonen, J.: Global assessment of shipping emissions in 2015 on a high
spatial and temporal resolution, Atmos. Environ., 167, 403–415,
https://doi.org/10.1016/j.atmosenv.2017.08.042, 2017.

Johnson, B.T., Shine, K.P. and Forster, P.M.: The semi-direct aerosol effect: Impact of absorbing aerosols
on marine stratocumulus, 1407–1422, https://doi.org/10.1256/qj.03.61, 2004.

Jordi, A., Basterretxea, G., Tovar-Sanchez, A., Alastuey, A. and Querol, X.: Copper aerosols inhibit
phytoplankton growth in the Mediterranean Sea, Proc. Natl. Acad. Sci., 109(52), 21246–21249,
https://doi.org/10.1073/pnas.1207567110, 2012.

Klein, S.A. and Hartmann, D.L.: The Seasonal Cycle of Low Stratiform Clouds, J. Clim., 6, 1587–1606,
https://doi.org/10.1175/1520-0442, 1993.

KPMG: Namibia Country mining guide: Strategy series, [online] Available from:
https://assets.kpmg.com/content/dam/kpmg/pdf/2014/09/namibia-mining-guide.pdf, 2014.

Kříbek, B., Šípková, A., Ettler, V., Mihaljevič, M., Majer, V., Knésl, I., Mapani, B., Penížek, V., Vaněk, A. and
Sracek, O.: Variability of the copper isotopic composition in soil and grass affected by mining and
smelting in Tsumeb, Namibia, Chem. Geol., 493, 121–135,
https://doi.org/10.1016/j.chemgeo.2018.05.035, 2018.

Lee, E., Chan, C.K. and Paatero, P.: Application of positive matrix factorization in source apportionment
of particulate pollutants in Hong Kong, Atmos. Environ., 33, 3201-3212,
https://doi.org/10.1016/S1352-2310(99)00113-2, 1999.

Lindesay, J.A., Andreae, M.O., Goldammer, J.G., Harris, G., Annegarn, H.J., Garstang, M., Scholes, R.J. and
van Wilgen, B.W.: International Geosphere-Biosphere Programme/International Global Atmospheric
Chemistry SAFARI-92 field experiment: Background and overview, J. Geophys. Res., 101, 23521–
23530, https://doi.org/10.1029/96JD01512, 1996.

Louw, D.C., Van Der Plas, A.K., Mohrholz, V., Wasmund, N., Junker, T. and Eggert, A. Seasonal and
interannual phytoplankton dynamics and forcing mechanisms in the Northern Benguela upwelling
system, J. Mar. Sys. Elsevier B.V., 157, 124–134, https://doi.org/10.1016/j.jmarsys.2016.01.009,
2016.

Lyyränen J., Jokiniemi J., Kauppinen E.I. and Joutsensaari J.: Aerosol characterisation in medium-speed
diesel engines operating with heavy fuel oils. J. Aerosol Sci. 30, 771–784. 10.1016/S0021-
8502(98)00763-0, 1999,

Maier, W. D., Rasmussen, B., Fletcher, I. R., Arnes, S.-J. and Huhma, H.: The Kunene Anorthosite Complex
, Namibia, and Its Satellite Intrusions: Geochemistry, Geochronology, and Economic Potential, Econ.
Geol., 108, 953–986, 2013.

Mänd, K., Kirsimäe, K., Lepland, A., Crosby, C.H., Bailey, J.V., Konhauser, K.O., Wirth, R., Schreiber, A. and
Lumiste, K.: Authigenesis of biomorphic apatite particles from Benguela upwelling zone sediments
off Namibia: The role of organic matter in sedimentary apatite nucleation and growth, Geobiology,
16, 640-658, https://doi.org/10.1111/gbi.12309, 2018.

Matengu, B., Xu, Y. and Tordiffe, E.: Hydrogeological characteristics of the Omaruru Delta Aquifer System
in Namibia, Hydrogeol. J., 27, 857–883, https://doi.org/10.1007/s10040-018-1913-0, 2019.

Morel, F.M.M. and Price, N.M.: The Biogeochemical Cycles of Trace Metals in the Oceans, Science, 300, 944–947, https://doi.org/10.1126/science.1083545, 2003.

Muhlbauer, A., McCoy, I.L. and Wood, R.: Climatology of stratocumulus cloud morphologies: Microphysical properties and radiative effects, Atmos. Chem. Phys., 14, 6695–6716, https://doi.org/10.5194/acp-14-6695-2014, 2014.

Namport: Annual Report, [online] Available at: https://www.namport.com.na/files/documents/dee_Annual%20Report%2012%20months%20en ded%2031%20March%202018.pdf; last access 19 February 2020.

Nelson, G., and Hutchings, L.: The Benguela upwelling area, Prog. Oceanog., 12(3), 333-356, https://doi.org/10.1016/0079-6611(83)90013-7, 1983.

Norris, G., Duvall, R., Brown, S. and Bai, S.: EPA Positive Matrix Factorization (PMF) 5.0 Fundamentals and User Guide. U.S. Environmental Protection Agency, Washington, DC, EPA/600/R-14/108 (NTIS PB2015-105147), 2014.

Okin, G.S., Baker, A.R., Tegen, I., Mahowald, N.M., Dentener, F.J., Duce, R.A., Galloway, J.N., Hunter, K., Kanakidou, M., Kubilay, N., Prospero, J.M., Sarin, M., Surapipith, V., Uematsu, M., and Zhu, T.: Impacts of atmospheric nutrient deposition on marine productivity: Roles of nitrogen, phosphorus, and iron, Global Biogeochemical Cycles, 25, 1-10, https://doi.org/10.1029/2010gb003858, 2011.

Paatero, P. and Tapper, U.: Positive Matrix Factorization: A non-negative factor model with optimal utilization of error estimates of data values, Environmetrics, 5, 111–126, https://doi.org/10.1002/env.3170050203, 1994.

Paatero, P., Eberly, S., Brown, S.G. and Norris, G.A.: Methods for estimating uncertainty in factor analytic solutions, 7, 781–797, https://doi.org/10.5194/amt-7-781-2014, 2014.

Painemal, D., Kato, S., Minnis, P., Funk, T., Hartmann, D.L., Short, D.A., Wilcox, E.M., Klein, S.A., Hartmann, D.L., Review, M.W., Chand, D., Wood, R., Anderson, T.L., Satheesh, S.K., Charlson, R.J., Costantino, L., Bréon, F.M., Systems, S., Systems, S., Systems, S., Darwin, E., Espy, J.P., Muhlbauer, A., McCoy, I.L., Wood, R., Medeiros, B., Zuidema, P., De Szoeke, S., Fairall, C. and Arakawa, A.: Aerosol indirect effect on warm clouds over South-East Atlantic, from co-located MODIS and CALIPSO observations, Atmos. Chem. Phys., 37, 6695–6716, https://doi.org/10.1175/1520-0469(1980)037, 2014a.

Painemal, D., Kato, S. and Minnis, P.: Boundary layer regulation in the southeast Atlantic cloud microphysics during the biomass burning season as seen by the A-train satellite constellation, J. Geophys. Res., 119, 11288-11302, https://doi.org/10.1002/2014JD022182, 2014b.

Painemal, D., Xu, K., Cheng, A., Minnis, P. and Palikonda, R.: Mean Structure and Diurnal Cycle of Southeast Atlantic Boundary Layer Clouds: Insights from Satellite Observations and Multiscale Modeling Framework Simulations, J. Clim., 28, 324–341, https://doi.org/10.1175/JCLI-D-14-00368.1, 2014c.

Paytan, A., Mackey, K.R.M., Chen, Y., Lima, I.D., Doney, S.C., Mahowald, N., Labiosa, R. and Post, A.F.: Toxicity of atmospheric aerosols on marine phytoplankton, PNAS, 106, 4601–4605, https://doi.org/10.1073/pnas.0811486106, 2009.

Preston-Whyte, R.A., Diab, R.D. and Tyson, P.D.: Towards an inversion climatology of Southern Africa: Part II, non-surface inversions in the lower atmosphere, South African Geogr. J., 59, 45–59, https://doi.org/10.1080/03736245.1977.9713494, 1977.

Prodi, F., Belosi, F., Contini, D., Santachiara, G., Di Matteo, L., Gambaro, A., Donateo, A., and Cesari, D.: Aerosol fine fraction in the Venice Lagoon: Particle composition and sources, Atmos. Res., 92, 141-150, https://doi.org/10.1016/j.atmosres.2008.09.020, 2009.

Quinn, P.K., Coffman, D.J., Kapustin, V.N., Bates, T.S. and Covert, D.S.: Aerosol optical properties in the marine boundary layer during the First Aerosol Characterization Experiment (ACE 1) and the underlying chemical and physical aerosol properties, J. Geophys. Res., 103, 16547-16563, 1998.

Ramadan, Z., Song, X.-H., Hopke, P.K.: Identification of sources of phoenix aerosol by positive matrix
factorization, J. Air. Waste. Manage., 50, 1308–1320,
https://doi.org/10.1080/10473289.2000.10464173. 2000.

Sanders, J. G. and Windom, H.L.: The uptake and reduction of arsenic species by marine algae, Estuar.
Coast. Mar. Sci., 10, 555–567, https://doi.org/10.1016/S0302-3524(80)80075-2, 1980.

Scheifinger, H. and Held, G.: Aerosol behaviour on the South African Highveld, Atmos. Environ., 31, 3497–
3509, https://doi.org/10.1016/S1352-2310(97)00217-3, 1997.

Seinfeld, J.H. and Pandis, S.N.: Atmospheric Chemistry and Physics: From Air Pollution to Climate
Change, 2nd ed., John Wiley & Sons, Hoboken, New Jersey., 2006.

Schembari, C., Bove, M.C., Cuccia, E., Cavalli, F., Hjorth, J., Massabò, D., Nava, S., Udisti, R., and Prati, P.:
Source apportionment of PM10 in the Western Mediterranean based on observations from a cruise
ship, Atmos. Environ., 98, 510-518, https://doi.org/10.1016/j.atmosenv.2014.09.015, 2014.

Šerbula, S.M., Antonijević, M.M., Milošević, N.M., Milić, S.M. and Ilić, A.A.: Concentrations of particulate
matter and arsenic in Bor (Serbia), J. Hazard. Mater., 181, 43–51,
https://doi.org/10.1016/j.jhazmat.2010.04.065, 2010.

Shibata, Y., Sekiguchi, M., Otsuki, A. and Morita, M.: Arsenic compounds in zoo- and phyto-plankton of
marine origin, Appl. Organomet. Chem., 10(9), 713–719, https://doi.org/10.1002/(SICI)1099-
0739(199611)10:9<713::AID-AOC536>3.0.CO;2-U, 1996.

Sinha, P., Hobbs, P.V., Yokelson, R.J., Christian, T.J., Kirchstetter, T.W. and Bruintjes, R.: Emissions of trace
gases and particles from two ships in the southern Atlantic Ocean, Atmos. Environ., 37, 2139–2148,
https://doi.org/10.1016/S1352-2310(03)00080-3, 2003.

Soto-Viruet, Y.: The Mineral Industries of Angola and Namibia, U.S. Geol. Surv., Minerals Y(November),
1–12 [online] Available from: https://www.usgs.gov/centers/nmic/africa-and-middle-east#na,
2015.

South African Weather Service (SAWS): Publications. [online] Available from:
http://www.weathersa.co.za/climate/publications. Date of access: 26 September 2020.

Smith, C. B.: Pb, Sr and Nd isotopic evidence for sources of southern African Cretaceous kimberlites,
Nature, 304, 51–54, https://doi.org/10.1017/CBO9781107415324.004, 1983.

Stein, A.F., Draxler, R.R., Rolph, G.D., Stunder, B.J.B., Cohen, M.D. and Ngan, F.: Noaa's hysplit atmospheric
transport and dispersion modeling system, Bull. Am. Meteorol. Soc., 96, 2059–2077,
https://doi.org/10.1175/BAMS-D-14-00110.1, 2015.

Swap, R., Garstang, M., Macko, S.A., Tyson, P.D., Maenhaut, W., Artaxo, P., Kållberg, P. and Talbot, R.: The
long-range transport of southern African aerosols to the tropical South Atlantic, J. Geophys. Res., 101,
23777–23791, https://doi.org/10.1029/95JD01049, 1996.

Swap, R.J., Annegarn, H.J., Suttles, J.T., King, M.D., Platnick, S., Privette, J.L. and Scholes, R.J.: Africa
burning: A thematic analysis of the Southern African Regional Science Initiative (SAFARI 2000), J.
Geophys. Res., 108, 8465, https://doi.org/10.1029/2003JD003747, 2003.

Taylor, M.P. and Kesterton, R.G.H.: Heavy metal contamination of an arid river environment: Gruben
River, Namibia, Geomorphology, 42(3–4), 311–327, https://doi.org/10.1016/S0169-
555X(01)00093-9, 2002.

Theobald, M.R., Crittenden, P.D., Hunt, A.P., Tang, Y.S., Dragosits, U. and Sutton, M.A.: Ammonia emissions
from a Cape fur seal colony, Cape Cross, Namibia, Geophys. Res. Lett., 33, 2–5,
https://doi.org/10.1029/2005GL024384, 2006.

Tlhalerwa, K., Freiman, M.T. and Piketh, S.J.: Aerosol Deposition off the Southern African West Coast by
Berg Winds, S. Afr. Geogr. J., 87, 152-161, https://doi.org/10.1080/03736245.2005.9713838, 2012.

Tournadre, J.: Anthropogenic pressure on the open ocean: The growth of ship traffic revealed by
altimeter data analysis, Geophys. Res. Lett., 41, 7924–7932,
https://doi.org/10.1002/2014GL061786, 2014.

Tyson, P.D., Garstang, M., Swap, R., Kållberg, P. and Edwards, M.: An air transport climatology for subtropical Southern Africa, Int. J. Climatol., 16, 265–291, https://doi.org/10.1002/(SICI)1097-0088(199603)16:3<265::AID-JOC8>3.0.CO;2-M, 1996.

Tyson, P.D. and Preston-Whyte, R.A.: The Weather and Climate of Southern Africa, 2nd ed. Oxford University Press Southern Africa, Cape Town, 2014.

Viana, M., Amato, F., Alastuey, A., Querol, X., Moreno, T., Dos Santos, S.G., Herce, M.D. and Fernández-Patier, R.: Chemical tracers of particulate emissions from commercial shipping, Environ. Sci. Technol., 43, 7472–7477, https://doi.org/10.1021/es901558t, 2009.

Vickery, K.J., Eckardt, F.D., Bryant, R.G.: A sub-basin scale dust plume source frequency inventory for southern Africa, 2005–2008. Geophys. Res. Lett., 40, 5274–5279, https://doi.org/10.1002/grl.50968, 2013.

Vouk, V.B. and Piver, W.T.: Metallic Elements in Fossil Fuel Combustion Products: Amounts and Form of Emissions and Evaluation of Carcinogenicity and Mutagenicity, Environ. Health Perspect., 47, 201–225, https://doi.org/10.1289/ehp.8347201, 1983.

Wanke, H., Nakwafila, A., Hamutoko, J.T., Lohe, C., Neumbo, F., Petrus, I., David, A., Beukes, H., Masule, N. and Quinger, M.: Hand dug wells in Namibia: An underestimated water source or a threat to human health?, J. Phys. Chem. Earth, 1–10, https://doi.org/10.1016/j.pce.2015.01.004, 2015.

Wanke, H., Ueland, J.S. and Hipondoka, M.H.T.: Spatial analysis of fluoride concentrations in drinking water and population at risk in Namibia, Water SA, 43, https://doi.org/10.4314/wsa.v43i3.06, 2017.

Wilcox, E. M.: Stratocumulus cloud thickening beneath layers of absorbing smoke aerosol, Atmos. Chem. Phys., 10, 11769–11777, https://doi.org/10.5194/acp-10-11769-2010, 2010.

Wood, R. Stratus and stratocumulus, in Encyclopedia of Atmospheric Sciences, 2nd ed., vol. 2, edited by G. R. North, J. Pyle, and F. Zhang, Elsevier, 196– 200, 2015.

Zhang, M., Chen, J.M., Wang, T., Cheng, T.T., Lin, L., Bhatia, R.S. and Hanvey, M.: Chemical characterization of aerosols over the Atlantic Ocean and the Pacific Ocean during two cruises in 2007 and 2008, J. Geophys. Res., 115, 1-15, https://doi.org/10.1029/2010JD014246, 2010.

Zorn, S.R., Drewnick, F., Schott, M., Hoffmann, T. and Borrmann, S.: Characterization of the South Atlantic marine boundary layer aerosol using an aerodyne aerosol mass spectrometer, Atmos. Chem. Phys., 8, 4711–472, https://doi.org/10.5194/acp-8-4711-2008, 2008.

Zuidema, P., Painemal, D., De Szoeke, S. and Fairall, C.: Stratocumulus cloud-top height estimates and their climatic implications, J. Clim., 22, 4652–4666, https://doi.org/10.1175/2009JCLI2708.1, 2009.

Zuidema, P., Redemann, J., Haywood, J., Wood, R., Piketh, S., Hipondoka, M. and Formenti, P.: Smoke and Clouds above the Southeast Atlantic: Upcoming Field Campaigns Probe Absorbing Aerosol's Impact on Climate, Bull. Am. Meteorol. Soc., 97, 1131-1135, https://doi.org/10.1175/BAMS-D-15-00082.1, 2016.

**Table captions**

**Table 1**. Summary statistics of elemental and water-soluble ionic concentrations measured at HBAO. The second column indicates the number of samples for which values were above the minimum quantification limit (MQL). The arithmetic means with standard deviations (sd) and range of mass concentrations (minimum and maximum) are given in ng m$^{-3}$.

**Table 2**. Annual arithmetic mean mass ratios of Cl$^-$, Mg$^{2+}$, K$^+$, Ca$^{2+}$, F$^-$ and SO$_4^{2-}$ with respect to Na$^+$ for 2016 and 2017. The Pearson coefficient of the lienar regression ($R^2$) is reported. Mass ratios for average seawater from Seinfeld and Pandis (2006) are shown for comparison. Standard deviations are indicated as *sd*.

**Table 3.** Annual arithmetic mean mass ratios of mineral dust tracers with respect to Al for 2016 and 2017. The Pearson coefficient of the lienar regression ($R^2$) is reported. Mass ratios for previous publications are shown for comparison. Standard deviations are indicated as *sd*.

[revised manuscript text omitted]

**Figure S2.** Gridded frequency plot of the variability of the 72-h air mass back-trajectories run for sampling periods in 2016 and 2017 (except the long-range transported air masses in September 2016 and November 2017, which show air masses arriving from as far south as 74°S) run for 21 of the 26 filter sampling periods. Back-trajectories were initiated at 250 m agl. The grid colour indicates the percentage of trajectories of the total trajectories run for all sampling periods, passing the over the 1° x 1° grid.

[Figure]

[Figure]

**Figure S4.** Bivariate polar plots for (a) vanadium and (b) nickel, showing the variability in mean concentrations with changes in wind speed and direction. Wind direction is indicated by the cardinal point in the four quadrants, mean wind speed (m s$^{-1}$) is indicated by the concentric circles from the centre of the plot and the mean concentrations are measured in ng m$^{-3}$, and given by the gradient colour scale.

(a)                                          (b)

[Figure]

**Figure S5.** Pie charts of the PMF mass apportionment of V and Ni measured at HBAO. Legends provide with the name of the source component and the fraction of the contributed mass elemental concentration.

[Figure]

**Figure S6.** Scatterplot of F⁻ with respect to nssCa²⁺ for 2016 (blue) and 2017 (orange). Concentrations are expressed in µg m⁻³. The slope and the Pearson correlation coefficient ($R^2$) are indicated.

[Figure]

**Figure S7**. Same as Figure S5 for F$_t$ concentrations.

[Figure]

